# HydroPy (v1.0): A new global hydrology model written in Python

Tobias Stacke[1] and Stefan Hagemann[1]

[1]Helmholtz-Zentrum Hereon, Institute of Coastal Research, Max-Planck-Strasse 1, 21502 Geesthacht, Germany

**Correspondence:** Tobias Stacke (tobias.stacke@hereon.de)

**Abstract.** Global hydrological models (GHMs) are a useful tool in the assessment of the land surface water balance. They are used to further the understanding of interactions between water balance components as well as their past evolution and potential future development under various scenarios. While GHMs are a part of the Hydrologist's toolbox since several decades, the models are continuously developed. In our study, we present the HydroPy model, a revised version of an established GHM, the Max-Planck Institute for Meteorology's Hydrology Model (MPI-HM). Being rewritten in Python, the new model requires much less effort in maintenance and due to its flexible infrastructure, new processes can be easily implemented. Besides providing a thorough documentation of the processes currently implemented in HydroPy, we demonstrate the skill of the model in simulating the land surface water balance. We find that evapotranspiration is reproduced realistically for the majority of the land surface but is underestimated in the tropics. The simulated river discharge correlates well with observations. Biases are evident for the annual accumulated discharge, however they can – at least to some part – be attributed to discrepancies between the meteorological model forcing data and the observations. Finally, we show that HydroPy performs very similar to MPI-HM and, thus, conclude the successful transition from MPI-HM to HydroPy.

*Copyright statement.* TEXT

## 1 Introduction

Hydrological models are used in a large number of research and operational applications. These range from the catchment scale e.g. highly calibrated models simulating discharge for one specific river, towards large temporal and spatial scales e.g. the global analysis of hydrological extremes in long-term climate projections (e.g., Do et al., 2020). Models employed in the latter type of studies belong to the group of global hydrology models (GHM), which focus on simulating different components of the hydrological cycle for the whole land surface.

Hydrological modeling looks back at several decades of model development (Bierkens, 2015). Starting from early land-surface schemes featuring a single bucket-style soil layer (Manabe, 1969), they grew in complexity over time implementing plant processes and multi-layer soil hydrology as well as considering human impacts (e.g. Alcamo et al., 1997). Depending on their area of operation, the models differ not only in terms of processes, but also in their technical infrastructure and programming language used. They can be designed for stand-alone applications using meteorological forcing data or be an interactively

coupled component in Earth System Models. Models with a long legacy are often written in Fortran (e.g. SWAT Arnold et al., 1998) or C (e.g. VIC Liang et al., 1994), but also other programming languages like Python (e.g. PCR-GLOBWB Sutanudjaja et al., 2018) are used. While models have come a long way since their initial concepts, there is still ample opportunity for improvement and extension of their functions. Bierkens (2015) identified issues due to missing geospatial data, calibration and input data quality, but also recognized the need to increase model resolution and eventually include sociological and economical interactions.

In order to face such challenges with our model, the Max-Planck Institute for Meteorology's Hydrology Model (MPI-HM; Hagemann and Dümenil, 1997; Stacke and Hagemann, 2012; Telteu et al., 2021), a thorough revision of its source code is required. The MPI-HM is a well established GHM, which was applied in several inter-comparisons projects and case studies over the last years (e.g., Haddeland et al., 2011; Wada et al., 2013; Schewe et al., 2013; Prudhomme et al., 2013; Mbaye et al., 2015; Zhao et al., 2017; Rasche et al., 2018; Hagemann et al., 2020; Pokhrel et al., 2021). Nonetheless, after a decade of demand-driven development, the MPI-HM became increasingly difficult to maintain. Furthermore, its source code – originally written in Fortran 77 style and later upgraded with Fortran 90 statements – made debugging and the implementation of new processes unnecessarily cumbersome. Therefore, it was decided to rewrite the model from scratch using Python, thus taking advantage of the high level routines provided by NumPy (Harris et al., 2020) and xarray (Hoyer and Hamman, 2017).

This revised version of MPI-HM – now renamed to HydroPy – uses the same hydrological process formulations and boundary data, but is easier to set up, more flexible in terms of spatial resolution, and offers a much higher potential for future enhancement. Consequently, a number of new processes were already implemented. The purpose of this article is to document the setup and structure of HydroPy, including both, the revised as well as newly implemented processes. Furthermore, HydroPy's ability to represent observed hydrological quantities is evaluated. Lastly, an overview of ongoing model development is presented and the challenges encountered during this project are discussed.

## 2 Model description

HydroPy (Stacke and Hagemann, 2021b) consists of a simplified land surface scheme and a lateral river routing module. It computes only water fluxes and does not feature an explicit energy balance. Instead, near surface air temperature is used as input for empirical formulations in energy related processes, e.g. snow melt or evaporation. If not mentioned otherwise, the near surface air temperature (or 2m temperature) is referred to as surface temperature throughout this study. Spatial resolution, time step, domain size, and simulation period are determined from the auxiliary data and meteorological forcing. The default setup, for which simulation data is presented in Sect. 4, uses a spatial resolution of 0.5° and a time step of 1 day.

Data describing land surface conditions and meteorological forcing are expected in SI units, e.g. [kg m$^{-2}$] for all water storages and [kg m$^{-2}$ s$-1$] for water fluxes. Internally, all fluxes are multiplied with the model time step $\Delta t$.

Each simulation starts with the computation of several secondary boundary fields derived from the primary input data (details in Sect. 3). As this is done only once per simulation, the numerical costs are low, but compared to the former MPI-HM the number of variables in the land surface data file is reduced from 29 to 18. Thus, new HydroPy users need to spend less

effort for compiling the required input data prior to model application. Next, the model loops over the time axis provided by the meteorological forcing data. In the beginning of the loop, surface temperature ($T_{srf}$), precipitation ($P$) and potential
evapotranspiration (PET) are read and the latter two are converted into mass flux per time step. Next, all land cover type (LCT) fractions are updated based on the actual date. This is followed by computing water fluxes for the various processes and updating the respective water storages. Figure 1 displays an overview about the water storages included in HydroPy and their connecting water flows. For most storages, outgoing water fluxes are computed based on the storage state at the start of the actual time step followed by updating the storage state using the actual fluxes. The order of storage updates is snow ➔ skin &
canopy ➔ soil ➔ surface water & groundwater ➔ river flow and the related processes are presented in detail in the paragraphs in section 2.2. At the end of every time step, all requested output variables are either written to a file (for time-step output) or added to an temporary field for time averaged output. Thus, no time series of variables need to be kept in memory, which strongly reduces the model's computational demands. All output variables are combined in one netCDF file per output time step resolution (daily, monthly, or yearly). Similar to the input, they are stored in SI units: [kg m$^{-2}$] for water storages, [kg
m$^{-2}$ s$^{-1}$] for water fluxes as well as [m$^3$] and [m$^3$ s$^{-1}$] for the special cases of river storage and river discharge, respectively.

At the end of the time loop, screen output is created which informs about the overall water balance and any residuum in the sum of water sources, sinks and storage changes. In case the water balance residuum exceeds a given threshold, an additional netCDF output file is generated which provides fields containing the water balance components. This acts as an early warning system during model development while implementing new processes. The water balance residuum is diagnosed for every grid
cell as one-dimensional water column and as water volume accumulated over the whole fields. Through the latter also water balance errors caused by changes in LCT fractions can be spotted easily.

In case of errors, there is an option to output time series for single grid cells in which all variables are included as well as partial water balances for the individual water storages at model time step. This supports the identification of processes responsible for a water balance violation.

## 2.1 Model design and setup

The rational behind HydroPy is not so much to generate a highly efficient GHM, but rather a model which can be easily understood by scientists and extended with additional processes. Thus, we decided to separate functions based on their purpose (see Fig. 2). The file `parameter_class.py` contains default options and parameter settings together with variable definitions, the `dataio_class.py` file handles all input and output functions required by HydroPy, the file `analysis_class.py`
includes functions related to spin-up, water balance checks and log data and the file `utility_routines.py` is a compilation of general routines like interpolation. All physical process representations are compiled in respective files describing their overall topic like hydrology, land cover and streamflow processes. This enables us to use the main routine to manage the simulation time loop and call the specific model processes in a clearly arranged manner. Furthermore, this file separation serves as a template for the implementation of extensions.
All variables representing physical properties are organised in a variable class that contains variable attributes and functions. Attributes comprise name, units and grid information. Functions are used to initialize the state of variables, average over time

periods and write output as well as restart data. At the start of the model one variable stream is created for all fluxes, states and land cover fractions, respectively. This allows to easily add new variables simply by adding some basic information to the respective definition function in the parameter file.

For the majority of hydrological processes, highly optimized NumPy arrays can be used because no information is exchanged between grid cells. However, this is different for river routing. Here, lateral water transport is considered (see Sect. 2.2.5), which requires iterations across the field indices. Such loops are rather slow in an interpreted programming language like Python. For this reason, the river routing routine was written not only in Python, but also in Fortran. Optionally, the Fortran code can be compiled into a shared library during model run-time using the NumPy tool `f2py` and called directly from HydroPy. During

model development, we made sure that identical results are returned from the Python and Fortran functions. Thus, the Python version can be used during model development and debugging while the Fortran based river routing is recommended for production simulations due to its higher efficiency (see Fig. 3).

The model itself can be called via command line and includes a `-help` option to display calling parameters and further options. A file containing meteorological forcing data is required. Default configurations can be changed by either providing a

105 json-style configuration file or by changing options directly via command line. A detailed technical documentation for model and simulation setup is part of the model repository (Stacke and Hagemann, 2021b).

Although the MPI-HM followed a similar idea in terms of separating processes in different files, the structure of HydroPy is much cleaner and easier to extend and modify due to the use of variable objects. Also, the included diagnostics like spin-up state evaluation and water balance computation as well as the interactive debugging options facilitate the implementation of

110 new processes. And finally, the simulation setup using a simple command line interface and a optional configuration file is a considerable improvement compared to the rather extensive run-scripts required for MPI-HM simulations.

## 2.2 Hydrological processes

The following paragraphs describe all processes implemented in HydroPy in detail and all variables used are summarized in Tab 1. Generally, the processes are identical to the MPI-HM in terms of equations. A few exceptions from this rule exist, which

are mentioned below. Furthermore, it should be noted that MPI-HM also includes a simple irrigation scheme which was used to explore the effect of human impacts on river discharge (e.g Haddeland et al., 2014). This scheme is not yet implemented in the current version of HydroPy, but will become part of a model version dedicated to land management in a future release.

### 2.2.1 Snow processes

In the snow module, the snow storage $S_{sn}$ is updated based on snowfall $P_{sn}$, snowmelt $R_{sn}$ and liquid snow storage content

$S_{snlq}$ as:

$$\Delta S_{sn} = P_{sn} - R_{sn} + \begin{cases} S_{snlq}, & \text{if } T_{srf} < 273.15 \text{ K} \\ 0 & \text{otherwise} \end{cases} \tag{1}$$

The involved water fluxes are computed as follows. Snowfall – if not provided directly in the forcing data – is diagnosed from precipitation as a function of surface temperature (Eq. 2) following Wigmosta et al. (1994). Snowmelt is calculated using the surface temperature $T_{srf}$ and day-length fraction $f_{lday}$ in a degree-day formular based on the HBV model (Bergström, 1992). Naturally, snowmelt is limited by the available snow storage.

$$P_{sn} = P \cdot \min\left\{1, \max\left\{0, \frac{T_{sn,max} - T_{srf}}{T_{sn,max} - T_{sn,min}}\right\}\right\}, \qquad \text{with } T_{sn,min} = 272.05 \text{ K and } T_{sn,max} = 276.45 \text{ K} \qquad (2)$$

$$R_{sn} = \min\left\{(f_{lday} \cdot 8.3 + 0.7) \cdot (T_{srf} - T_{melt}), S_{sn} + P_{sn}\right\}, \qquad \text{with } T_{melt} = 273.15 \text{ K} \qquad (3)$$

While MPI-HM used a sine function to estimate the daylength fraction based on the day of the year, this is replaced in HydroPy by explicitly computing the day-length based on the Earth's declination (Forsythe et al., 1995). This has the advantage that a reasonable snowmelt is computed also for the southern hemisphere. However, the effects are only visible for very few grid cells due to the small extent of snow affected areas in the southern hemisphere.

Part of the snowmelt can be retained ($F_{snlq}$) as liquid snow storage content and is either subject to refreezing – if surface temperatures fall below the freezing point – or sustains snowmelt in the next time step. The liquid snow storage content is limited to a maximum of 6% of the total snow storage, which is a common value also used in other models (e.g. Wigmosta et al., 1994).

$$F_{snlq} = S_{sn} \cdot f_{snlq,max} - S_{snlq}, \qquad \text{with } f_{snlq,max} = 0.06 \qquad (4)$$

$$R_{sn} = R_{sn} - F_{snlq} \qquad (5)$$

$$S_{snlq} = S_{snlq} + F_{snlq} \qquad (6)$$

### 2.2.2 Skin and Canopy processes

A new addition to HydroPy is the implementation of a skin $S_{skin}$ and canopy storage $S_{can}$, which enhances the evaporation $E$ for the grid cell fractions with the LCTs bare soil $f_{bare}$ and vegetated $f_{veg}$, respectively. Both storages receive input from rainfall $P_{ra}$ and snowmelt $R_{sn}$ and provide overflowing water as throughfall towards the ground. Evaporation scales linearly with PET and available water, but is additionally limited by the maximum storage content $S_{prc,max}$. Mechanistically, both storages are treated equally and are described using the index $prc$ to denote the different $skin$ und $can$ processes:

$$\Delta S_{prc} = (P_{ra} + R_{sn} - E_{prc}) \cdot f_{lct}, \text{ with } prc = \{\text{skin, can}\} \text{ and } lct = \{\text{bare, veg}\} \qquad (7)$$

The related fluxes are defined for the respective fraction of the grid cell and scaled to the overall grid cell during the final balancing. Thus, for all storage interactions its content needs to be normalized to the fraction $f_{lct}$ resulting in $S_{prc,norm} = \frac{S_{prc}}{f_{lct}}$ so that:

$$f_{wet} = \min\left\{1, \frac{S_{prc,norm} + P_{ra} + R_{sn}}{S_{prc,max}}\right\} \qquad (8)$$

$$E_{prc} = \text{PET} \cdot f_{wet} \cdot \min\left\{1, \frac{S_{prc,norm} + P_{ra} + R_{sn}}{f_{wet} \cdot \text{PET}}\right\} \qquad (9)$$

$$R_{prc} = \max\left\{0, S_{prc,norm} + (P_{ra} + R_{sn} - E_{prc}) - S_{prc,max}\right\} \qquad (10)$$

While the processes for skin and canopy storage are parameterized similarly, they use specific maximum storage values:

$$\Delta S_{prc,max} = cap_0 \cdot \begin{cases} 1, & \text{if } prc = \text{skin} \\ \text{LAI} & \text{if } prc = \text{can} \end{cases}, \text{with } cap_0 = 0.2 \tag{11}$$

Both runoff fluxes are scaled by the respective LCT fractions and combined to provide the throughfall $R_{tr}$ that reaches the soil surface:

$$R_{tr} = R_{skin} \cdot f_{bare} + R_{can} \cdot f_{veg} \tag{12}$$

### 2.2.3 Soil processes

HydroPy employs a single-layer soil scheme in which the storage $S_{so}$ provides water for transpiration $E_T$, bare soil evaporation $E_{bs}$, and drainage $R_{gr}$ (used synonymously for groundwater recharge or subsurface runoff) towards a shallow groundwater storage. Furthermore, its content determines the surface runoff $R_{srf}$. The soil storage does not represent the physical soil over a given depth but only the water column stored in the soil. Thus, its maximum value is limited by the maximum soil water capacity $S_{so,max}$. The soil moisture storage is defined as:

$$\Delta S_{so} = R_{tr} - R_{srf} - R_{gr} - E_T \cdot f_{veg} - E_{bs} \cdot f_{bare} \tag{13}$$

Both, drainage and surface runoff, are implemented identically to these fluxes in MPI-HM using the same parameter values. Drainage, following the formulation by Dümenil and Todini (1992), is defined such that it scales linearly with soil moisture content up to a given threshold. Above, saturated flow is assumed resulting in accelerated downward percolation with exponential scaling:

$$R_{gr,low} = R_{gr,min} \cdot \Delta t \cdot \frac{S_{so}}{S_{so,max}} \tag{14}$$

$$R_{gr,high} = (R_{gr,max} - R_{gr,min}) \cdot \Delta t \cdot \left( \frac{S_{so} - S_{so,grmax}}{S_{so,max} - S_{so,grmax}} \right)^{R_{gr,exp}}, \qquad \text{with } R_{gr,exp} = 1.5 \tag{15}$$

Both are combined as:

$$R_{gr} = \begin{cases} 0, & \text{if } S_{so} \leq S_{so,grmin} \text{ or } T_{srf} < 273.15 \text{ K} \\ R_{gr,low} & \text{if } S_{so,grmin} < S_{so} \leq S_{so,grmax} \\ R_{gr,low} + R_{gr,high} & \text{if } S_{so,grmax} < S_{so} \end{cases} \tag{16}$$

The soil moisture thresholds $S_{so,grmin}$ and $S_{so,grmax}$ are products of $S_{so,max}$ using the factors 0.05 and 0.9, respectively. The minimum and maximum drainage, $R_{gr,min}$ and $R_{gr,max}$ are set to $2.7e-7$ and $2.7e-5$ [kg m$^{-2}$ s$^{-1}$]. As both parameters are given as flux per second, they need to be multiplied with the model time step $\Delta t$. All drainage related model parameters are taken from Roeckner et al. (1992).

The surface runoff $R_{srf}$ representation follows the Improved Arno Scheme (Hagemann and Gates, 2003). It assumes that $S_{so,max}$ is not homogeneous for a whole grid cell, but varies on sub-grid scale. Thus, parts of the grid cell, where the local

storage capacity is low, can already generate surface runoff even though the cell average soil moisture state is still below its average maximum moisture holding capacity. Therefore, a fraction of $R_{tr}$ is converted into $R_{srf}$ as soon as the minimum soil moisture content is exceeded. This is realized by mapping $S_{so}$ onto the sub-grid soil moisture capacity distribution parameters denoted by the index $sg$:

$$S_{so,sg} = \begin{cases} S_{so,sg,max} - (S_{so,sg,max} - S_{so,sg,min}) \cdot \left(1 - \frac{S_{so} - S_{so,sg,min}}{S_{so,max} - S_{so,sg,min}}\right)^{\frac{1}{1+b}} & \text{if } S_{so} > S_{so,sg,min} \\ S_{so} & \text{otherwise} \end{cases} \tag{17}$$

where $b$ is a shape parameter describing the distribution of sub-grid soil moisture capacities within a grid cell (see Sect. 3). Overall $R_{srf}$ is computed as:

$$R_{srf} = \begin{cases} R_{tr} & \text{if } T_{srf} < 273.15 \text{ K} \\ 0 & \text{if } S_{so,sg} + R_{tr} \leq S_{so,sg,min} \text{ or } R_{tr} = 0 \\ R_{tr} + \max\{0, S_{so} - S_{so,max}\} & \text{if } S_{so,sg} + R_{tr} > S_{so,sg,max} \\ R_{tr} - \max\{0, S_{so,sg,min} - S_{so}\} - \frac{S_{so,sg,max} - S_{so,sg,min}}{1+b} \cdot (c_1 - c_2) & \text{otherwise} \end{cases} \tag{18}$$

where $c_1$ and $c_2$ are further scaling operations put into separate equations for convenience:

$$c_1 = \min\left\{1, \left(\frac{S_{so,sg,max} - S_{so,sg}}{S_{so,sg,max} - S_{so,sg,min}}\right)^{1+b}\right\} \tag{19}$$

$$c_2 = \max\left\{0, \left(\frac{S_{so,sg,max} - S_{so,sg} - R_{tr}}{S_{so,sg,max} - S_{so,sg,min}}\right)^{1+b}\right\} \tag{20}$$

The subgrid parameters $S_{so,sg,max}$ and $S_{so,sg,min}$ that determine the range of subgrid maximum water capacity variability are spatially distributed fields and their generation is discussed in Sect. 3.

As for runoff, transpiration $E_T$ and bare soil evaporation $E_{bs}$ are implemented based on the respective MPI-HM equations (which themselves are derived from Roeckner et al., 1996; Warrilow et al., 1986; Bauer et al., 1983) and use a linear scaling with soil moisture content. Even though they are attributed to specific LCT fractions, the vegetated $f_{veg}$ and bare soil $f_{bare}$ parts of the grid cell, they don't rely on fraction specific soil moisture storages but use the grid cell average for simplicity. Still, it has to be considered that part of the evaporative demand on these fractions is already satisfied due to canopy and skin evaporation:

$$E_T = (\text{PET} - E_{can}) \cdot \min\left\{1, \max\left\{0, \frac{S_{so} - S_{so,wilt}}{f_{so,crit} \cdot S_{so,max} - S_{so,wilt}}\right\}\right\}, \qquad \text{with } f_{so,crit} = 0.75 \tag{21}$$

$$E_{bs} = (\text{PET} - E_{skin}) \cdot \min\left\{1, \max\left\{0, \frac{S_{so} - f_{so,bs,low} \cdot S_{so,max}}{(1 - f_{so,bs,low}) \cdot S_{so,max}}\right\}\right\}, \qquad \text{with } f_{so,bs,low} = 0.05 \tag{22}$$

Again, the spatially distributed data fields $S_{so,max}$ and $S_{so,wilt}$ are part of the input dataset (Sect. 3).

Runoff and evaporation fluxes are visualized in Fig. 4 for a single grid cell. There, the threshold dependent change in the scaling of individual fluxes in respect to specific thresholds can be easily recognized.

### 2.2.4 Surface water and shallow groundwater processes

The implementation of surface water has been modified compared to the formulation used in MPI-HM and being described in Stacke and Hagemann (2012). In MPI-HM the vertical land surface water balance module and the river routing module were strongly separated, each featuring storages which were restricted to be used within the respective module. The sole exception was the surface water storage resulting from ponding water on the land surface, and, hence, used to represent small creeks, lakes and wetlands. This storage was part of the vertical land surface water balance but could also interact with the lateral river routing scheme. In HydroPy, we simplify the representation of lakes and wetlands and utilize storages that are already part of the routing scheme as will be explained in the next paragraphs.

The river routing scheme used in both, MPI-HM and HydroPy, is the Hydrological Discharge Model (HD-Model; Hagemann and Dümenil, 1997; Hagemann et al., 2020). To represent river flow, it employs three storages: the overland flow storage receives the local grid cell's surface runoff, the baseflow storage receives the local drainage and the river flow storage receives inflow from upstream grid cells. All three storages provide water to the next downstream grid cell. The purpose of the overland and baseflow storages is to delay the flow of both runoff components before they enter the river. As such, they represent a major function of surface water (e.g. small lakes, wetlands and creeks smaller than the grid cell size) and shallow groundwater - even though no effort was taken to explicitly represent any other hydrological functions.

In HydroPy, the former overland and baseflow storages are used as surface water $S_{sw}$ and groundwater storages $S_{gw}$. However, we like to emphasize that both storage representations are very limited and are not expected to simulate all relevant lake and groundwater processes. Neither circulation within lakes nor confined aquifers, fossil groundwater or lateral water movement within large scale aquifers are considered at this point. The surface water balance is adapted to allow for interception of rainfall and snow melt as well as water loss due to evaporation over the surface fraction $f_{sw}$ while the shallow groundwater balance only considers groundwater recharge $R_{gr}$ and runoff $R_{gw}$:

$$\Delta S_{sw} = (P_{ra} + R_{sn} - \text{PET}) \cdot f_{sw} + R_{srf} - R_{sw}, \text{ where } f_{sw} = \max\{f_{lake}, f_{wetland}\} \tag{23}$$
$$\Delta S_{gw} = R_{gr} - R_{gw} \tag{24}$$

Surface water evaporation is set to PET as long as there is enough water available over the surface water fraction. Surface water and groundwater runoff are both computed as:

$$R_{prc} = S_{prc} \cdot \frac{1}{\text{LAG}_{prc} + 1}, \text{ with } prc = \{\text{sw,gw}\} \tag{25}$$

The storage retention time $\text{LAG}_{prc}$ is computed based on the inner grid cell slope and grid cell size. More details can be found in Sect. 3.

### 2.2.5 River routing

River routing is realized based on formulations from the HD-Model, which was also part of the MPI-HM. It applies a river flow cascade with up to five intermediate storages ($c_{max}$), parametrizations for river flow retention times and uses prescribed

routing directions. It is the only part of HydroPy that uses a volumetric unit [m$^3$] instead of water columns. Thus, runoff needs to be converted using the grid cell area $A_{gc}$. The river flow storages $S_{rf,n}$ for the individual cascade members $n$ are defined as:

$$\Delta S_{rf,n} = Q_{rf,n-1} - Q_{rf,n} \qquad\qquad \text{with } n = \{1,..,c_{max}\} \text{ where} \qquad\qquad (26)$$

$$Q_{rf,n} = (S_{rf,n} + Q_{rf,n-1}) \cdot \frac{1}{\text{LAG}_{rf} + 1} \qquad\qquad (27)$$

$$Q_{rf,0} = \sum Q_{rf,up} \qquad\qquad (28)$$

$$Q_{rf} = Q_{rf,c_{max}} + (R_{sw} + R_{gw}) \cdot 0.001 \cdot A_{gc} \qquad\qquad (29)$$

where the index $up$ denotes the river flow $Q_{rf}$ from upstream grid cells. High river flow velocities combined with a daily model time step can lead to numerical instabilities at a given resolution in case water would travel more than one grid cell. For this reason, the river flow cascade as well as the routing are encapsulated in a sub-step time loop which is called several times per model time step.

## 3  Land surface data

In HydroPy an effort was made to reduce the number of land surface datasets to a minimum. Thus, very specialized datasets like the river flow lag times are computed by HydroPy itself instead of requiring the model user to pre-process such data. Of course, this comes with the disadvantage of a slight increase in computational demand, but as these data fields are only computed once per simulation, this is hardly noticeable. All land surface data fields are collected in one parameter file (Stacke and Hagemann, 2021a).

### 3.1  Primary data fields

All computations in HydroPy are restricted to land cells which are defined in the land sea mask as values $> 0$. All other fields providing fractional information on LCTs are normalized with the land sea mask and therefore represent relative fractions. The LCTs considered comprise glaciers, permafrost, lakes, wetlands and vegetation. Glacier cells (Hagemann, 2002) are omitted from the simulation if the glaciers cover the full land surface fraction while permafrost (GEWEX ISLSCP Project, 2007) strongly reduces the soils capability to store water. The lake and wetland fractions are based on the Global Lake and Wetland Database (GLWD; Lehner and Döll, 2004). They are utilized to scale surface water evaporation, surface water runoff and modify lateral flow velocities. The lake fractions are used without any modifications, but wetlands were restricted to the classes floodplains and peatlands for this study. Thus, we can best resemble the general wetland distribution used for the predecessor of HydroPy and facilitate a clean comparison between both (see Sect. 4.4).. The vegetated fraction is taken from the Land Surface Parameters Dataset 2 (LSP2; Hagemann, 2002) and included into the HydroPy parameter file as monthly mean climatology. Another vegetation parameter dataset use by HydroPy is the monthly LAI climatology, which also originates from the LSP2 dataset. For both, the monthly mean values are linearly interpolated to match the respective model time step. The vegetation fraction affects canopy processes and transpiration while the LAI defines the size of the canopy storage.

Three topographical variables are included in the parameter file, which are all based on the ETOPO1 dataset (Amante and Eakins, 2009). These variables are the grid cell mean surface orography and its standard deviation as well as the average slope within a grid cell. The slope was initially computed at the native dataset resolution ($1' \approx 0.01°$). Mean slope, mean orography and its standard deviation were determined during upscaling from the original to the target resolution of 0.5°. Both, mean orography and slope, are used to derive lateral water flow velocities in surface water, groundwater and rivers. The orographical standard deviation plays a role in determining the surface runoff.

The majority of the remaining parameters are related to soil properties. Maximum soil moisture capacity and plant available water are computed following the procedure described in Hagemann et al. (1999), but with a different dataset as source for plant available water. Values for plant available water are derived via a multi-linear regression from a dataset of optimized plant rooting depth (Kleidon, 2004) and high resolution land cover data (Loveland et al., 2000). The same data was also used in the MPI-ESM, the Earth system model of the Max-Planck Institute for Meteorology Hagemann and Stacke (2015) . While the maximum soil moisture capacity $S_{so,max}$ affects all soil processes, the plant available water $S_{so,wava}$ is needed to calculate transpiration. Further properties are the sub-grid minimum and maximum soil moisture capacity as well as an exponent $b_{sg}$ that describes the distribution of sub-grid soil moisture capacities within every model grid cell. All three of these values are discussed in Hagemann and Gates (2003) and are needed to compute surface runoff.

The final two parameters describe the river routing network using routing target indices. There is one field for each geographical dimension, i.e. latitude and longitude, and each grid cell is assigned with a flow target in the respective direction, which must be one of the eight neighboring grid cells. Note, that the target is provided as grid cell index, not as geographical coordinate. As the indices follow Python notation, they start at 0. For cells without any valid target flow target, its own indices are used. The directions prescribed with the routing target indices for HydroPy are identical to the one used by MPI-HM. After their first publication in Hagemann and Dümenil (1997), they were continuously improved during the last years.

## 3.2 Derived data fields

Not all land surface variables are used directly in the simulated processes. Instead, they are (also) used to derive other variables. This is done during model initialization and will be discussed in the following.

The impact of permafrost on the simulation is represented purely from a hydrological view point. All soil moisture capacity variables are reduced to reflect that only a small part of the soil column is not frozen and may actively contribute to the water balance. The reduction factor $f_{pe,red}$ is computed based on the permafrost cover fraction $f_{pe}$ as

$$f_{pe,red} = \frac{S_{pe,max} \cdot f_{pe} + S_{so,max} \cdot (1 - f_{pe})}{S_{so,max}} \text{ with } S_{pe,max} = 50 \text{ kg m}^{-2} \tag{30}$$

and then used to scale all soil moisture capacity fields:

$$S = S \cdot f_{pe,red} \text{ with } S = \{S_{so,max}, S_{so,wilt}, S_{so,sg,min}, S_{so,sg,max}\} \tag{31}$$

Next, the shape parameter $b$ describing the sub-grid distribution of soil moisture capacities (see Eqs. 15, 19, and 20) is derived based on the distribution parameter $b_{sg}$ and modified with the normalized orographical standard deviation $b_{oro}$:

$$b_{oro} = \frac{\sigma_h - \sigma_0}{\sigma_h + \sigma_{max}}, \qquad\qquad \text{where } \sigma_0 = 100 \text{ m and } \sigma_{max} = 1000 \text{ m} \tag{32}$$

$$b = \begin{cases} b_{sg} + b_{oro} & \text{if } b_{oro} > 0.01 \\ b_{sg} & \text{otherwise} \end{cases} \tag{33}$$

where $\sigma_h$ is the standard deviation of subgrid orography while the parameter $\sigma_0$ and $\sigma_{max}$ need to be modified according to the models spatial resolution.

Finally, the retention times LAG are determined, which are used in the simulation of surface water and shallow groundwater

flow (Eq. 25) as well as river flow (Eq. 27). In all three cases, reference values obtained by sensitivity experiments (Table 3, Hagemann and Dümenil, 1997) for a representative grid cell are modified using the actual properties of grid cells for the remaining part of the land surface – specifically a reference lag time, the flow distance $\Delta l$ and the flow velocity $v$. The flow velocity is computed based on the slope $s$ as:

$$v = \max\{v_{min}, c \cdot s^{\alpha}\}, \text{ with } c = 2, \alpha = 0.1 \tag{34}$$

with the minimum flow velocity set to 0.1 [m s$^{-1}$]. The actual lag for surface water $sw$ and river flow $rf$ is obtained by modifying the reference lag time (see Table 3) for these processes $prc$:

$$LAG_{prc} = LAG_{prc,ref} \cdot \frac{\Delta l_{prc}}{\Delta l_{prc,ref}} \cdot \frac{v_{prc,ref}}{v}, \text{ with } prc = \{sw, rf\} \tag{35}$$

As surface water flow is limited to the respective grid cells, the flow distance $\Delta l_{sw}$ is set to the cell's diameter and the slope to the average slope within the grid cell. In contrast, river flow transports water from one cell to the next. Thus, the flow

distance $\Delta l_{rf}$ is measured as the distance between the centers of a grid cell and its downstream grid cell. Similarly, the slope is computed based on the height difference $\Delta h$ between actual and downstream grid cell as $s = \frac{\Delta h}{\Delta l_{rf}}$. Compared to surface water and river flow, shallow groundwater flow is assumed to be more independent of surface slope and the reference lag is modified using the grid cell diameter as flow distance and the normalized variation in orography $b_{oro}$:

$$LAG_{gw} = LAG_{gw,ref} \cdot \frac{\Delta l_{gw}}{\Delta l_{gw,ref}} \cdot \frac{1}{1 - b_{oro} + 0.01} \tag{36}$$

Additionally, the maximum number of cascade storages $c_{max}$ (see Eq. 26) is assigned to surface water and river properties. The reference values are listed in table 3. The lag times of surface water and river flow are further modified if the lake or wetland fractions are large enough to interact with lateral water flow as described in Hagemann and Dümenil (1998). Here, it is assumed that a small water fraction hardly impacts lateral flow velocities but slows down flow velocity significantly if a certain threshold $f_{lct,crit} = 0.5$ is crossed. This behaviour is reflected in the following equations, where the flow reduction $f_{lct,red}$ is

computed:

$$f_{lct,red} = 0.5 \cdot (\tanh(4 \cdot \pi \cdot (f_{lct} - f_{lct,crit})) + 1) \text{ where } lct = \{\text{lake, wetland}\} \tag{37}$$

The flow velocity of both processes $prc$ is normalized with reference flow velocities $v_{lct,ref}$ for for the land cover types ($lct$) lakes and wetlands – set to 0.01 and 0.06 m s$^{-1}$, respectively – and scaled with the flow reduction:

$$v_{prc} = \frac{\Delta l_{prc}}{\text{LAG}_{prc} \cdot c_{max,prc} \cdot \Delta t}, \text{ with } prc = \{sw, rf\} \tag{38}$$

$$v_{prc,red} = v_{prc} \cdot \left(1 - \left(1 - \frac{v_{prc}}{v_{lct,ref}} \cdot f_{lct,red}\right)\right) \tag{39}$$

The number of storages within the flow cascade and the lag time are then updated accordingly:

$$c_{max,prc} = (c_{max,prc} - 1) \cdot (1 - f_{lct,red}) + 1 \tag{40}$$

$$\text{LAG}_{prc} = \frac{\Delta l_{prc}}{v_{prc,red} \cdot c_{max,prc} \cdot \Delta t} \tag{41}$$

These computations are done for both water storage types $prc = \{$surface water,river$\}$ and both land cover types $lct = \{$lake,wetland$\}$. As a final step, all retention times are scaled such that the maximum cascade storage number $c_{max}$ becomes an integer value and can easily be applied in an iteration.

## 4   Evaluation

The HydroPy evaluation is based on a global simulation, set up at 0.5°spatial resolution with a time step of 1 day. Precipitation and surface temperature time series were taken directly from the Global Soil Wetness Project Phase 3 forcing data (GSWP3, Dirmeyer et al., 2006), while potential evaporation was computed from the same dataset in a pre-processing step using the Penman-Monteith reference evaporation approach (Allen et al., 1998). The simulation period was initialized with a spin-up run over 50 iterations of the year 1979 and then run until 2014. The last 30 years were used for the evaluation and comparison against different observational datasets.

### 4.1   Simulation spin-up

While it is possible to convert restart files from MPI-HM to initialize HydroPy, the much cleaner way is to start with empty storages and run the model until the trends in its storages become reasonable small. With the currently implemented processes, a time period of 50 years was considered to be sufficient for this task.

The upper left map in Fig. 5 shows the total water storage (TWS) trends during the first 10 years of the spin-up simulation with values up to $100$ kg m$^{-2}$ a$^{-1}$. The largest trends are found within the tropical zone as well as in the south-eastern monsoon region due to the large amount of precipitation together with relative deep soils that provide a large potential storage. The TWS trends for the last 10 years of spin-up (Fig. 5, upper right map) are below $1$ kg m$^{-2}$ a$^{-1}$ for the vast majority of grid cells except some single cells dominated by glaciers (see below).

Analysing the individual water storages of the spin-up simulation (see Fig. 5, blue lines), their simulated contents averaged over the land surface become stable within 10 years or less, with the exception of snow water equivalent where single grid cells continuously accumulate snow. These cells are located close to the coasts of Greenland and Antarctica. They are not masked

as their glacier fraction is smaller than the land fraction, but they have surface temperatures below $0°C$ for most of the year. Currently, no glacier flow processes are implemented in HydroPy and therefore these cells cannot reach a stable state. However, this issue affects less then $2\%$ of the snow-covered grid cells and these cells can easily be masked during the analysis.

The remaining storages differ in the number of years needed to reach a stable state. Small, highly dynamical storages like canopy and skin (not shown) are well initialized already during the first year, while storages with larger values and slower processes like soil moisture, groundwater and river storage show positive trends in their mean state until year 8. Thus, 50 years of spin-up seems somewhat excessive at a first glance. However, the mean storage state is not necessarily representative for the whole land surface as can be demonstrated by considering the storage content for the individual grid cells that show the largest residual trend at the end of the 50 year spin-up period (Fig. 5, orange lines). In the most extreme case, soil moisture spin-up takes the major part of the 50 years period for a grid cell in the Ganges delta where the soil moisture capacity is much larger than for the majority of the other cells. Another effect is seen in the groundwater storage of a cell in the same region where a stable state was almost achieved after 40 years but was followed by a second increase period. This behaviour results from the drainage calculation (Eq. 16) where an exponential component is added in case the soil moisture exceeds a given threshold. Thus, groundwater spin-up in cells with a positive water balance and large soil moisture storage cannot be completed until the overlying soil reaches an stable state. For river storage, the most extreme cell is not spinning up much slower than the average, but looking at the single cell data, the storage content is shown to exceed those of other cells by several orders of magnitude. This cell is located at the eastern border of Lake Ontario where the main path of the St. Lawrence River crosses a grid cell with a lake fraction only slightly smaller than one. This results in a strongly reduced flow velocity in this cell and consequently a huge water storage is build up. In summary, a shorter period of 10 years would be sufficient for most of the land surface. However, regional features very well justify a spin-up period of 50 years. Of course, for later studies the length of spin-up simulations can be significantly reduced using restart data from another suitable HydroPy run.

## 4.2 Evaporation

As precipitation is provided as forcing, the major task of HydroPy is to realistically distribute precipitation into evaporation and runoff. Thus, a good representation of evaporation is crucial to successfully simulate river discharge. Unfortunately, there are no global datasets of directly observed evaporation time series available that could be used for model validation. However, data is available from the Global Land Evaporation Amsterdam Model (GLEAM; Martens et al., 2017; Miralles et al., 2011; ICDC, 2019), a sophisticated modelling system dedicated to estimate evaporative fluxes based on satellite observation and reanalysis data.

Comparing the long term zonal mean actual evaporation between HydroPy and GLEAM (Fig. 6, upper panel) both datasets agree very well for the high and mid latitudes in terms of zonal mean and standard deviation. However, for the tropical region between $10°$ and $-15°N$ HydroPy underestimates evapotranspiration by almost $1 \, \mathrm{kg \, m^{-2} \, d^{-1}}$. Considering seasonal differences (Fig. 6, lower panel), some more deviations become evident. During MAM the underestimation of tropical evapotranspiration is most pronounced affecting a band of more than $10°$ width south of the equator. During JJA this is compensated by a slight

overestimation south of -10°N while the evaporation underestimation shifts northwards. Also a small ($\approx 0.5 \, \mathrm{kg} \, \mathrm{m}^{-2} \, \mathrm{d}^{-1}$)
negative bias is visible between 30° and 70°N.

While the implication of these biases for runoff are discussed in a following paragraph (see Sect. 4.3), we aim to identify possible reasons for their appearance. The most obvious source would be potential evaporation, which is provided to HydroPy as input based on atmospheric forcing data and used in every evaporation computation as maximum value – especially as the Penman-Monteith methodology is shown to underestimate PET in tropical regions (Weiland et al., 2015). However, PET is overestimated compared to GLEAM (see Fig. 7) and thus unlikely to cause too low values in ET. Instead, the ET biases are well reflected in transpiration with even stronger values. This could hint on either issues with the overall size of the soil moisture storage or indicate a missing interaction between the shallow groundwater and soil moisture storages which could provide plants with additional water for transpiration during dry periods. The importance of a deep soil moisture layer for transpiration was already demonstrated by Hagemann and Stacke (2015) and might be relevant for HydroPy as well.

## 4.3 River discharge

River discharge is the most important output variable of HydroPy. It is an easily observable quantity that integrates the lateral water fluxes over a given catchment region and it is highly sensitive to all other components of the hydrological cycle. As such, river discharge is a prime target to evaluate the skill of GHMs. This is done for the whole globe using observations provided by GRDC (2020).

We use the Kling-Gupta Efficiency (KGE, Gupta et al., 2009) as primary evaluation metric for river discharge. The KGE is defined as:

$$KGE = 1 - \sqrt{(r-1)^2 + \left(\frac{\sigma_{sim}}{\sigma_{obs}} - 1\right)^2 + \left(\frac{\mu_{sim}}{\mu_{obs}} - 1\right)^2} \tag{42}$$

with $r$ being the correlation between simulated ($sim$) and observed ($obs$) river discharge time series, $\sigma$ being their standard deviation and $\mu$ being their long-term mean value. The KGE ranges in the bounds between $\{-\infty, 1\}$, with a value of $KGE = -0.41$ indicting a performance similar to using the observed mean flow (Knoben et al., 2019). The open lower bound complicates the comparison of KGE values between different simulations or models as their differences can become very large for catchments with unsatisfactory model performance, while changes in well capture catchments are restricted to a much smaller range. For this reason, we use a normalization when comparing KGE values following the approach used by Nossent and Bauwens (2012) for the Nash-Sutcliffe Efficiency (NSE) remapping its value range to $\{0, 1\}$. We compute the difference of normalized KGE values as:

$$\Delta NKGE = \frac{1}{2 - KGE_{sim_{exp}}} - \frac{1}{2 - KGE_{sim_{ref}}} \tag{43}$$

for all KGE comparisons.

Please note, that river discharge observations are often not available directly at the river mouth. For our analysis, we use those stations which are located at the most downstream part of the catchment and compare it with discharge from the closest grid cell with similar inflow area. Nonetheless, the resulting values are projected onto the whole catchment for visualization.

Additionally, we analyse the temporal correlation $r$ and the percentile bias $pb$ given by the equations:

$$r = \frac{\sum \left(Q_m - \overline{Q_m}\right)\left(Q_o - \overline{Q_o}\right)}{\sqrt{\sum \left(Q_m - \overline{Q_m}\right)^2 \sum \left(Q_o - \overline{Q_o}\right)^2}} \tag{44}$$

$$pb = \frac{\sum_{t=1}^{T}\left(Q_m^t - Q_o^t\right)\cdot 100}{\sum_{t=1}^{T} Q_o^t} \tag{45}$$

Figure 8 (histogram and upper map) shows the KGE for simulated and observed data of monthly river discharge climatolo-
gies in the 100 largest catchments of the world. Only about 25% of the catchment covered area shows a KGE $\leq -0.41$, which
indicates an insufficient performance as just using the observed long-term average is superior to the prediction. Thus, HydroPy
shows a reasonable performance for the majority of catchments. About 65% of the catchment area exceed a KGE of 0 and 33%
even exceed a KGE of 0.5. Contrary to the NSE, which was often used in hydrological analysis (Moriasi et al., 2007) prior to
the development of the KGE, the individual KGE values are not associated with a specific performance rating. Still, positive
values are generally considered to indicate skilful model application (Knoben et al., 2019). Taking into account that HydroPy
is neither calibrated nor adapted to any specific catchment, its performance is satisfactory for the majority of catchments and
on the same level as its predecessor MPI-HM (see Sect. 4.4).

Still, some river catchments could not be represented sufficiently well. Thus, additional metrics were calculated to identify
reasons for this failure. Figure 8 (lower left panel) shows the temporal correlation between HydroPy and GRDC monthly
mean river discharge, with a good performance for most catchments. More than two third of the grid cell show a correlation
$> 0.75$ and less then 15% are below 0.5. Thus, HydroPy demonstrates a significant skill to represent seasonal river discharge
variations in the majority of catchments. Exceptions can be seen for the Nile and Ganges rivers which also show low KGE
values. Stronger regional discrepancies are evident for the percentile bias between HydroPy and GRDC river discharge (Fig.
8, lower right panel). Only 20% of the grid cells fall in the range of $\pm 10\%$ bias. For high northern latitude catchments, a
strong underestimation of river discharge is visible, especially for Eastern Siberia and north-western Canada. Strong discharge
overestimation is found for many dry regions in the Sahel zone and Australia, but also in parts of South America. Combining
these measures (Fig. 8, upper right panel) shows that low KGE values occur mostly in regions with strong, positive biases in
mean discharge amounts and, hence, in total runoff. Therefore, this needs to be a focus of future model development. However,
it might be possible that not all of this issues can be addressed by the model alone. Figure 9 shows the long-term water
balance residuum based on GSWP3 precipitation, GLEAM evaporation and GRDC river discharge. While it is only a first
order approximation using a mixture of observed and computed quantities, it also highlights large negative values for Eastern
Siberia and north-western Canada as well as positive values for parts of South America, the Sahel Zone and the Ganges basin.
Especially the wet bias areas coincide well with the regions showing strong river discharge biases in HydroPy. This points to
the possibility that already the forcing or validation data might suffer from uncertainties and, thus, contribute to the diagnosed
river discharge biases for the these regions.

For regional river discharge curves (see Fig. 10), it can be seen that for Arctic Rivers the peak flow month is represented
very well, but the amount of peak flow is underestimated even though ET is too low during summer. In terms of model skill,
this could be related to too little accumulated snow cover during the winter months. Also during August and September, the

discharge stays below the observed level. Whether this is an effect of too small water storages in the soil, biases in the model forcing or related to other processes needs to be examined in later studies. For tropical rivers, a too early flow peak combined with too much discharge is visible. This can be attributed mostly to the strong negative ET bias during MAM and might be also related to the missing representation of interactive floodplains which might enhance ET and mitigate the peak flow. However, it should be noted that the discharge curve is strongly dominated by Amazon and Congo, while other tropical rivers, like Orinoco, Mekong and Rio Magdalena, are much better simulated. Similarly, river discharges for mid-latitude regions like the European Rivers are represented very well by HydroPy.

## 4.4 Comparison to MPI-HM

In this section, HydroPy is compared to the original MPI-HM to prove its suitability as a successor. For this purpose, a simulation was conducted with MPI-HM using the same meteorological forcing data and spin-up duration. Similar to HydroPy, MPI-HM includes a number of grid cells with steadily increasing snow storage. These cells were masked during the analysis of both simulations.

Figure 11 shows the average water fluxes and storage contents for HydroPy and MPI-HM. As expected, both are rather similar. The average precipitation received by the models amount to a yearly sum of $\approx 760 \, \mathrm{kg \, m^{-2} \, a^{-1}}$. Evapotranspiration in HydroPy exceeds the one in MPI-HM by $\approx 20 \, \mathrm{kg \, m^{-2} \, a^{-1}}$ while total runoff is $\approx 11 \, \mathrm{kg \, m^{-2} \, a^{-1}}$ lower. The soil moisture storage is around $\approx 12 \, \mathrm{kg \, m^{-2}}$ higher in MPI-HM compared to HydroPy. Although both models agree in the spatial distribution of soil moisture trends, the drying trends are slightly stronger in the HydroPy simulation resulting in an overall lower soil moisture. The average snow water equivalent is similar in both models with a difference of $< 1 \, \mathrm{kg \, m^{-2}}$. Still, regional differences can be found featuring an increase in snow cover in the high northern latitudes (due to later snow melt) balanced by a decrease in mid-latitudes. These small differences can be explained by the new processes implemented in HydroPy. The addition of skin and canopy evaporation increases overall evapotranspiration in HydroPy resulting in a slight decrease in soil moisture and therefore an decrease in total runoff. The differences in SWE are caused by the replacement of the normalized sinus function in the snow melt scheme with an explicit calculation of the day-length fraction.

Comparing the quality of simulated river discharge between both models (see Fig. 12), HydroPy shows slightly better results in the normalized Kling-Gupta efficiency (NKGE) for about $60\%$ of the area of considered catchments. While the changes in NKGE values are mostly small, there are some catchments with stronger signals, e.g. the Ganges-Brahmaputra, Yangtze and Yenisey. Climatologies for the three rivers are shown in the upper panels of Fig. 13. For these basins, HydroPy generates a delay in river flow compared to MPI-HM and GRDC observations, which results in a later peak flow by 1 to 2 months and an overall smoother curve. This behaviour results from the representation of surface water in HydroPy and its effect on river flow velocity. Indeed, the wetland masks used in both models differ. MPI-HM relied on a wetland distribution compiled by Matthews and Fung (1987). However, this dataset is only available at the rather low resolution of $1°$. In order to enable HydroPy to run on resolutions higher than the default $0.5°$ in future applications, lakes and wetlands are now compiled using the GLWD (Lehner and Döll, 2004, see Sect. 3), which is available at 5' resolution. While both maps agree on the general distribution of wetlands, there are regional differences. Most prominently, the Matthews and Fung (1987) dataset contains less wetlands than the GLWD

in the Ganges Delta, the downstream area of the Yangtze River and the outlet of the Yenisey, which results in the overestimation of wetland impacts in HydroPy. The same effect causes the increase in NKGE for the Mississippi and Yukon catchments (see Fig. 13, middle panels) where HydroPy reproduces the timing of the peak flow much better than MPI-HM although it falls off too quickly during the summer in the case of the Yukon. Here, the wetland impact acts too strong in MPI-HM while the selection of GLWD wetland classes in HydroPy leads to less wetlands in this area. Another example demonstrating the small differences between HydroPy and MPI-HM is the improved river discharge in the Danube catchment. Here, the small changes in the snow scheme slightly reduce the snow cover and thereby the snow melt peak.

The river discharge simulation for Ganges-Brahamaputra, Yangtze and Yenisey can be significantly improved in HydroPy by adapting the reference flow velocity for wetlands (see Fig. 13, lower panel). However, the effect is diametrical to the skill in other wetland influenced catchments. An increase in the reference velocity reduces the wetland impact, thereby improving the simulation skill towards the level of MPI-HM. At the same time, it results in performance degradation in other catchments like the Ob, where due to the faster river flow through wetlands the snow melt peak appears earlier in the year while too less water is available during late summer.

Besides exploring the reasons behind differences between HydroPy and MPI-HM, this analysis demonstrates the strong sensitivity of river discharge to wetland extent in both models. However, the opposing effects of modifying the wetland reference flow velocities discourages any further attempts to optimize this value at this point. More likely, better results can be obtained at higher resolution when flow paths and wetland distribution are represented in greater detail. Then, only those wetlands would mitigate river discharge which are indeed close enough to the flow channel and not simply exist in the same 0.5°grid cell.

Finally, it should be noted that due to the active participation of MPI-HM in several inter-comparison projects (e.g. Haddeland et al., 2011; Zhang et al., 2017; Gädeke et al., 2020; Kumar et al., 2021), there are many analyses available that evaluate the performance of MPI-HM in comparison to other GHMs. As MPI-HM and HydroPy produce very similar results, these analyses are also a good indication for the performance of HydroPy at this stage of development.

## 5 Conclusions

This study introduces HydroPy, a global hydrological model written in Python. Similar to its predecessor, the MPI-HM, the model simulates the land surface water balance as well as the river routing towards the ocean. HydroPy uses a default resolution of 0.5° on global scale. It has rather humble data requirements, using surface temperature, precipitation and potential evaporation as meteorological input data together with land surface property data. Compared to MPI-HM, HydroPy can be setup more easily, has less requirements for land surface data, provides internal checks to aid the model development process and requires less effort for maintenance and implementation of new processes.

The majority of process formulations were taken over from MPI-HM without any alterations, but some improvements were done for the snow melt, the skin and canopy storages as well as the impact of lakes and wetlands on river discharge. The evaluation of evapotranspiration using the GLEAM data (Martens et al., 2017; Miralles et al., 2011; ICDC, 2019) shows a generally reasonable behaviour but also highlights underestimation of transpiration in parts of the Tropics and the high northern

latitudes. This affects runoff generation and results in an overestimation of river discharge especially for the Amazon and Congo rivers compared to GRDC data (GRDC, 2020) and also a too early discharge peak for these two rivers. The seasonality of Arctic rivers is captured very well but the peak discharge is generally too low. Although constraint to specific regions, both biases are considered to be important deficiencies – either in the model or in the meteorological forcing data – and will be a focus of the next development steps. Still, the majority of catchments show a very good correlation with observations and overall discharge can be satisfactorily represented by HydroPy.

Besides the necessary improvements in these parametrizations, there are further plans to actively develop HydroPy in the future. The two most important focus areas are the

- transport of nutrients within the lateral flow scheme of HydroPy to provide ocean models not only with river discharge, but also transient information about nutrient concentration,

- implementation of human impacts, especially irrigation and crop management, but also the representation of dams and reservoirs.

Compared to MPI-HM, HydroPy shows very similar results on the global scale. However, there are regional differences due to changes in the snow melt scheme and the updated wetland distribution. While these changes cause a degraded performance of river discharge in a few river basins, the majority of rivers benefit from the improvements implemented in HydroPy. Nonetheless, regions with decreased simulation quality will be in the focus of further model development. In conclusion, the transition from MPI-HM to HydroPy is completed successfully and HydroPy will replace MPI-HM in future applications.

As final remarks we present some insights gained during this project. Naturally, HydroPy is not the first hydrology model to be rewritten (e.g. SWAT+ Bieger et al. (2016) and others), however the difficulties and challenges of such a project are not often discussed (Müller et al., 2018). A major obstacle is the perception of this work: rewriting a model prepares the way for new studies but is often considered a technical rather than a scientific task. For this reason, it is difficult to secure funding, computational resources, and finally publish this work in high-ranking journals. As scientists are often employed on limited contracts, they need to carefully weight such time investments because model development, maintenance and documentation usually pays off only on long time scales. For this reason, small working groups without technical support are at a disadvantage when trying to rewrite their models. Still, in terms of the technical realization, we like to share some advice that was useful to us:

- Programming language and style should be chosen according to the user base and application, e.g. Fortran or C for highly-optimized models running on high-performance machines vs interpreted languages like Python or R with easily understandable code, e.g. for educational purposes or working groups without a technical support staff.

- Infrastructural routines should be separated from the scientific code as much as possible. If I/O and variable definitions happen "under the hood", the main routine can be much more easily understood and extended by other scientists. For this very reason, we recommend to develop the model infrastructure as the very first step and only later add the scientific processes.

- Implement automated checks right from the beginning. Standardized computations of water/mass balances and sensible additional outputs, e.g. log outputs for selected grid cells, help to spot any conceptual or programming errors as early as possible.

- Port one routine at a time and summarize related functions in one file for a better overview. A modular model can be much more easily extended for different applications.

- Write documentation right from the start and not only when the first model version is finished.

- Use a versioning tool (git, svn) to manage the development progress, collaborations and releases.

Of course, all this advice comes from personal preferences and other programmers might set different priorities. Nonetheless, we hope it might ease the start on such projects because ultimately, a well structured and maintained open-source model is a valuable asset in hydrological research.

*Code availability.* The source code of HydroPy is available under the GNU General Public License v3.0 at Zenodo (https://doi.org/10.5281/zenodo.4541380)

*Data availability.* The land surface property data used for the presented HydroPy simulations is available under the Creative Commons Attribution-NonCommercial-NoDerivatives 4.0 International Public License (CC BY-NC-ND 4.0) at Zenodo (https://doi.org/10.5281/zenodo.4541238). The raw data produced with HydroPy and MPI-HM is stored at the CERA database (https://doi.org/10.26050/WDCC/HydroPy_MPI-HM_hist_sim) and and published under the Creative Commons Attribution-ShareAlike 4.0 International Public License (CC BY-SA 4.0).

*Author contributions.* TS developed the HydroPy model and conducted the experiments for this study. SH prepared the original land surface data, developed the land surface and river routing models which later became the MPI-HM, and provided input during the development of HydroPy. Both authors participated in the design of the study, the analysis of the simulation as well as in the writing and editing of the manuscript.

*Competing interests.* The authors declare that they have no conflict of interest.

*Acknowledgements.* TS is supported by the DAAD (#57429828) from funds of the German Federal Ministry of Education and Research (BMBF). The HydroPy logo (Fig. 1, upper right) was created by Raphael Dlugosch and Maria Lyssenko with contributions by Tobias Stacke and Richard Stacke. We like to thank two anonymous reviewers for the constructive comments that helped to improve our manuscript.

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

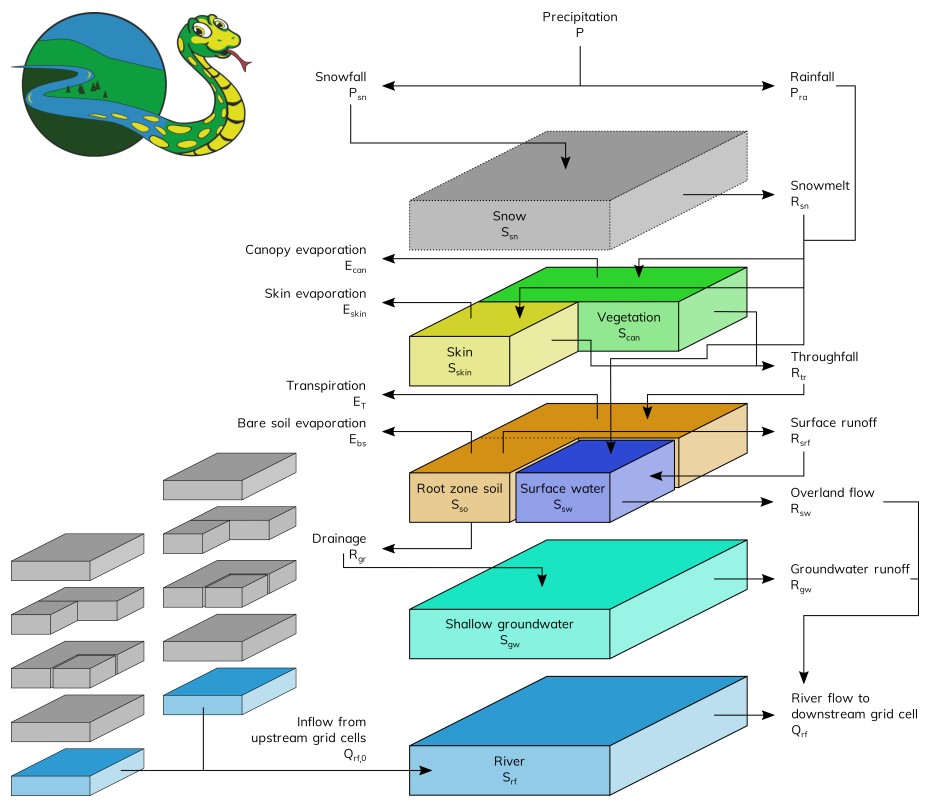

**Figure 1.** Schematic of the major water storages and fluxes in HydroPy together with the symbols used in the process description (see Sec. 2.2) Most processes only affect each grid cell individually (central column) but river routing also interacts with neighboring cells (represented by small storage columns) The upper left figure shows the official logo designed for the HydroPy model.

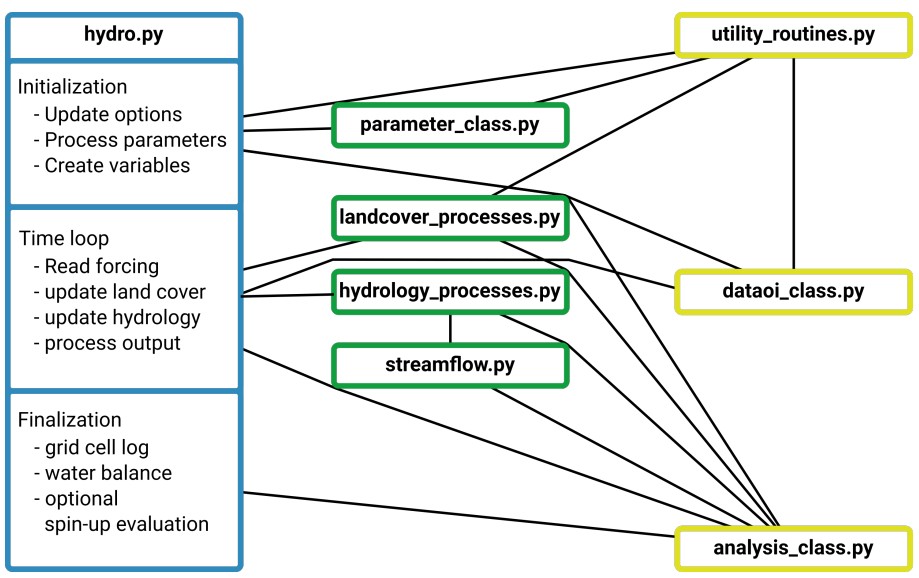

**Figure 2.** Simplified flowchart of HydroPy depicting the files belonging to the model and their primary interactions. The main routine is shown in blue, process routines are shown in green and files containing auxiliary functions (I/O, analysis, ...) are shown in yellow.

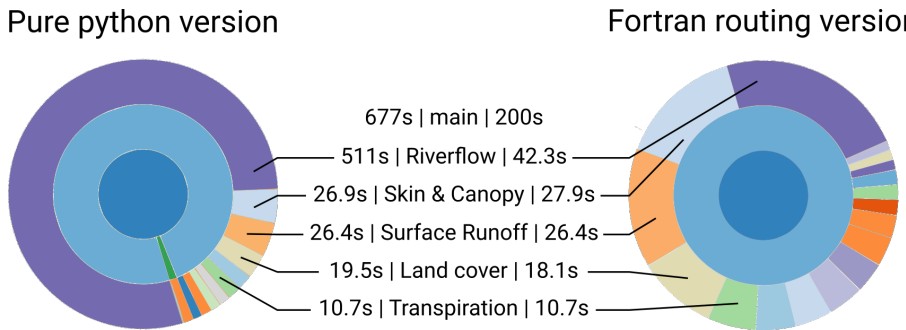

**Figure 3.** Sunburst graphs depicting the distribution of model run-time for a global simulation of one year at 0.5°resolution without writing any output. Both simulations were conducted on a single node of a high-performance Linux cluster (using a Intel(R) Xeon(R) Platinum 8160 CPU with 2.10GHz). Values are shown for the pure Python version (left) and the version using Fortran for river routing (right). The visualization was done using Snakeviz (https://jiffyclub.github.io/snakeviz/). Note, that as HydroPy does currently not utilize parallelization, the run-times on a standard office laptop are roughly the same.

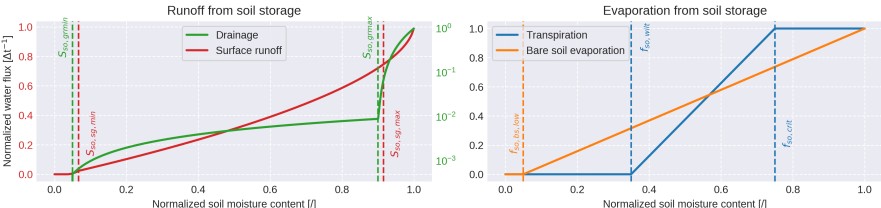

**Figure 4.** Behaviour of runoff (left panel, surface runoff in red, drainage in green) and evaporation (right panel, transpiration in blue, bare soil evaporation in orange) fluxes calculated as a function of soil moisture for a single grid cell. All variables are normalized using their maximum values for better comparison. Dashed lines indicate variable specific thresholds with are defined in section 2.2.3. Note, that drainage is visualized using a logarithmic axis.

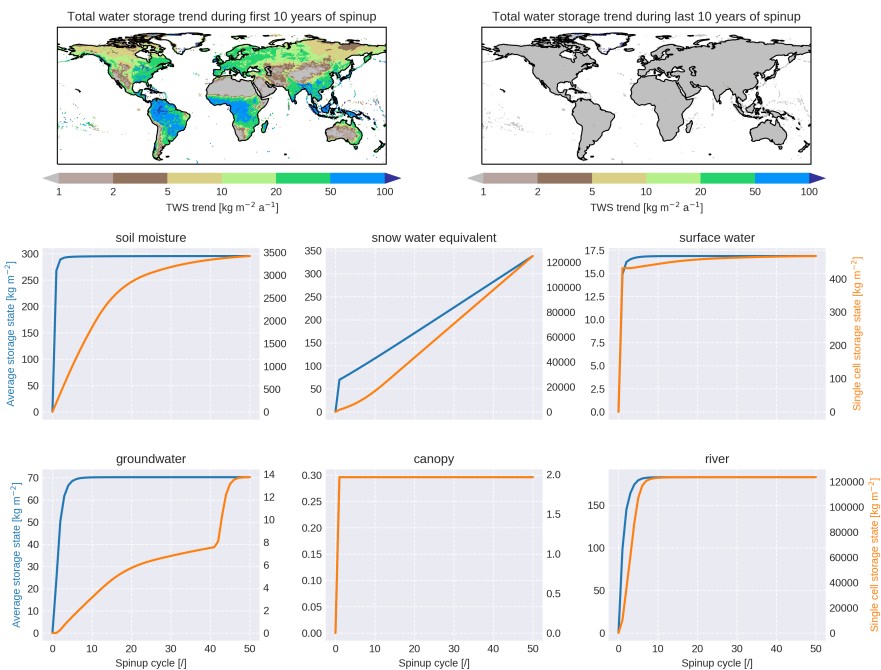

**Figure 5.** Evolution of various storage contents during HydroPy spin-up. The maps shows the trends in the total water storage (all storages accumulated) during the first (left map) and last (right map) 10 years of spin-up. Blue lines show values averaged over the whole land surface and orange lines represent the grid cell with the largest residual trend at the end of the spin-up period.

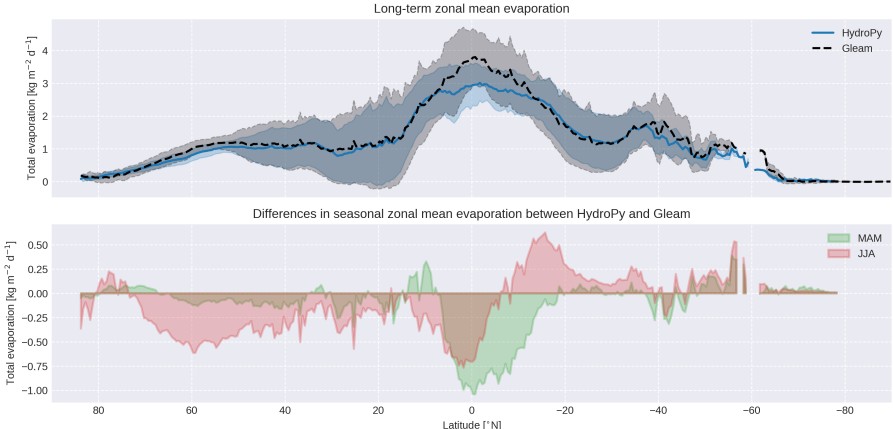

**Figure 6.** Zonal mean evaporation averaged over the period 1980-2014 for HydroPy and Gleam (upper panel) as well as their seasonal differences in the boreal spring (MAM) and summer (JJA) (lower panel).

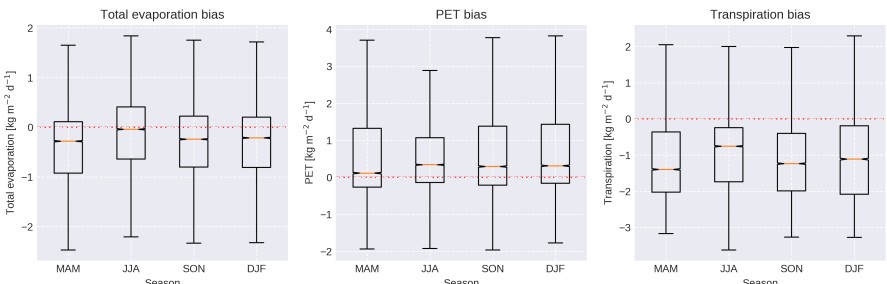

**Figure 7.** Percentile distribution of biases in the differences of seasonal mean fields between HydroPy and Gleam data between 20 and -20°N.

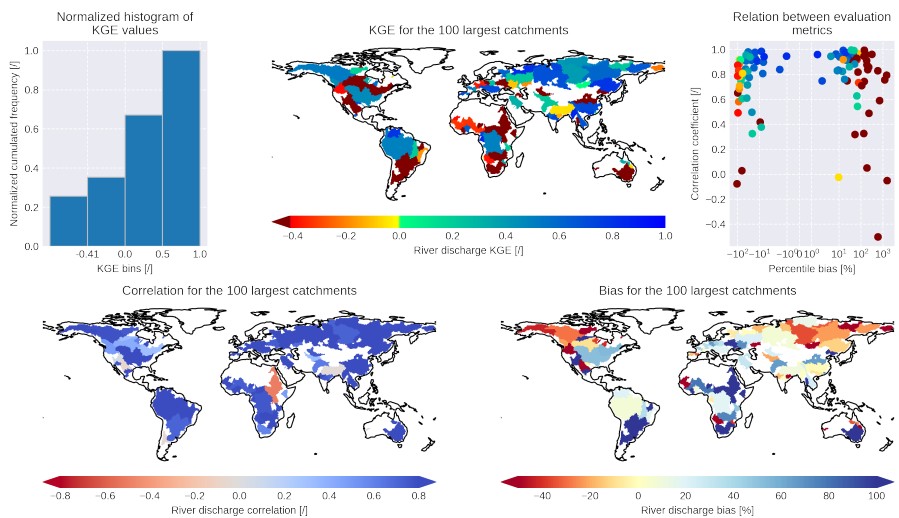

**Figure 8.** Kling-Gupta efficiency (KGE) of simulated river discharge climatologies for the world's 100 largest catchments compared to GRDC observations (GRDC, 2020): the upper panel shows the cumulative histogram (left) of the spatial distribution (middle) of KGE values as well as its relation (right) to the temporal correlation and percentile bias. The spatial distribution of the two latter metrics are shown in the lower panels.

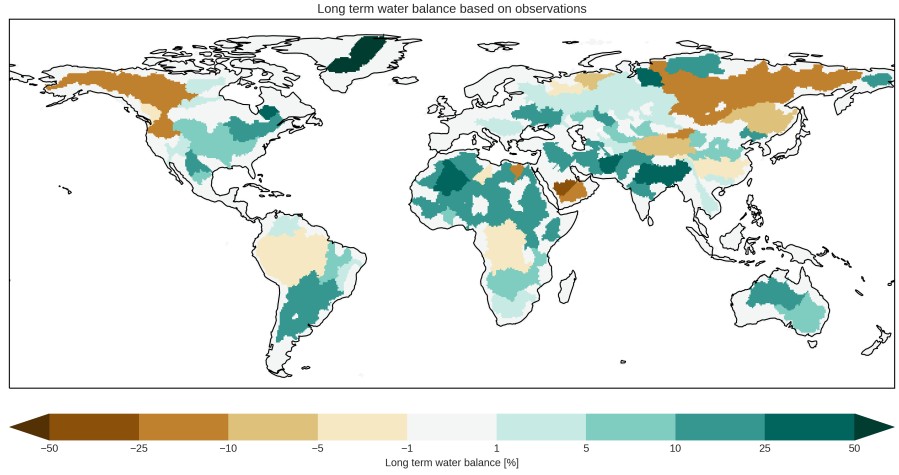

**Figure 9.** Long-term water balance based on GSWP3 precipitation, GLEAM evaporation and GRDC river discharge remapped to catchment scale. Note, that this water balance is only a first order estimate. No trends in storage content are included. Furthermore, the GRDC river discharge climatologies often represent different time periods than GSPW3 and GLEAM.

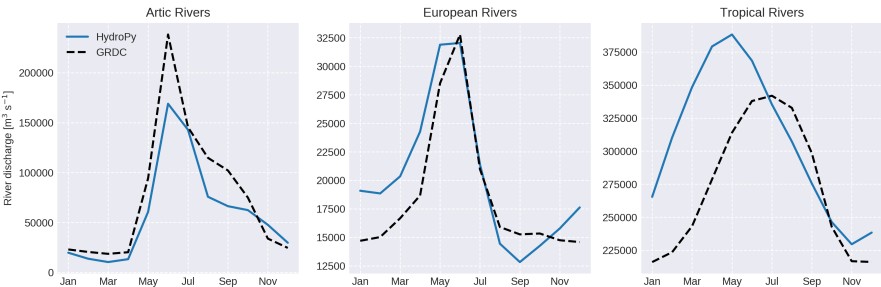

**Figure 10.** Climatological river discharge curves averaged for selected rivers in three regions: Arctic Rivers (Lena, Ob, Yenisey, Amur, Kolyma, Mackenzie River, and Yukon), European Rivers (Danube, Rhine, Rhone, Po, Vistula, Pechora, and Neva) and Tropical rivers (Amazon, Congo, Orinoco, Mekong, Rio Magdalena, Zambezi, and Niger). Simulated HydroPy discharge curves are indicated in blue and GRDC observations in dashed black.

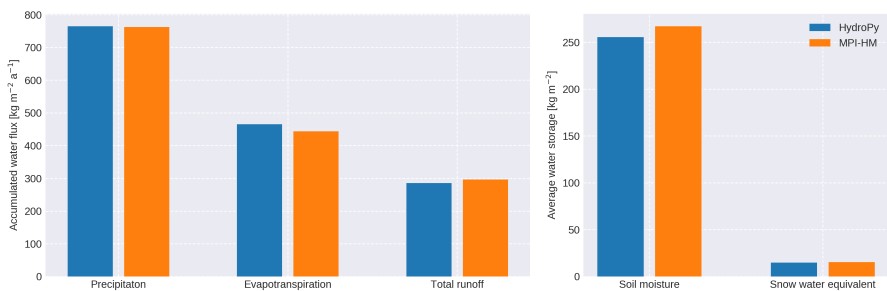

**Figure 11.** Land surface water fluxes (left) and storages states (right) averaged over the simulation period for HydroPy (blue) and MPI-HM (orange).

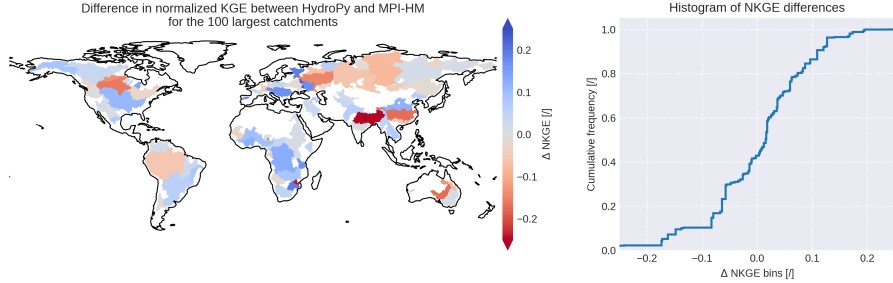

**Figure 12.** Spatial distribution (left) and cumulative histogram (right) of NKGE differences for river discharge between HydroPy and MPI-HM.

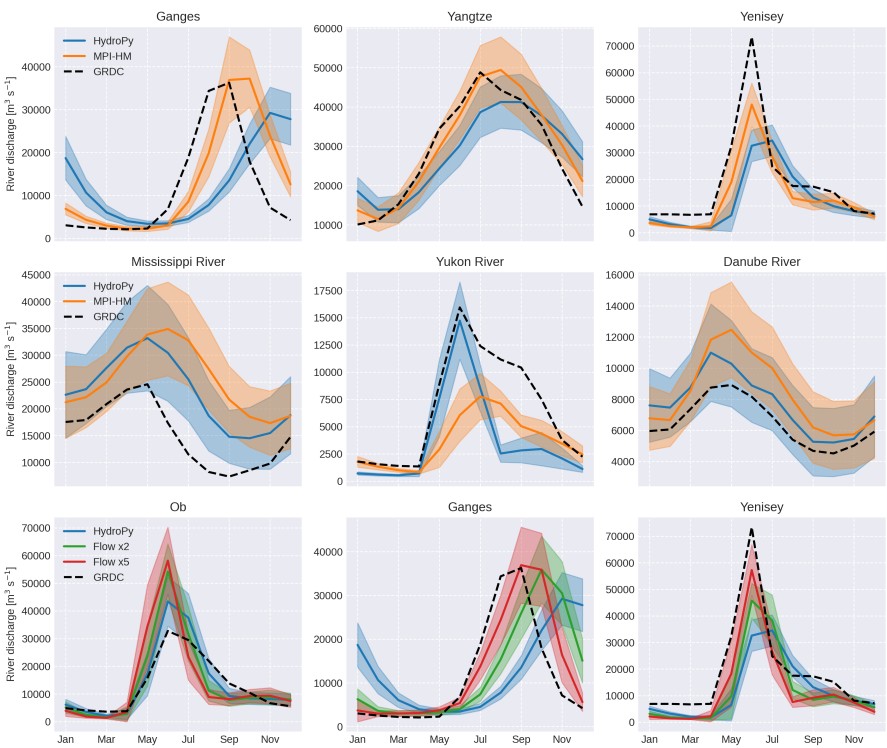

**Figure 13.** Monthly climatologies of river discharge for different river catchments with GRDC observations (dashed black). The upper two rows also show simulated discharge from HydroPy (blue) and MPI-HM (orange). The lower row show simulated river discharge produced with the default parameters in HydroPy (blue) as well as those with modified reference flow velocities for wetlands using factors of two (green) and five (red).

**Table 1.** Overview about variables used in HydroPy process description, referenced with their equations numbers. Variables directly available for output are denoted by providing their short name in the output column.

| Symbol | Description | Output Name | Equation |
|--------|-------------|-------------|----------|
| $A_{gc}$ | Grid cell surface area | - | 29 |
| $b$ | Orographical shape parameter | - | 17, 18, 19, 20, 33 |
| $b_{oro}$ | Normalized topographical standard deviation | - | 32, 36 |
| $b_{sg}$ | Subgrid distribution parameter | - | 32 |
| $c_{max}$ | Maximum number of river flow cascades | - | 26, 38, 40, 41 |
| $E_T$ | Plant transpiration | transp | 13, 21 |
| $E_{bs}$ | Bare soil evaporation | sevap | 13, 22 |
| $E_{can}$ | Canopy evaporation | canoevap | 7, 9, 10, 21 |
| $E_{skin}$ | Skin evaporation | skinevap | 7, 9, 10, 22 |
| $f_{bare}$ | Bare soil fraction | fbare | 7, 12, 13 |
| $f_{lake}$ | Lake fraction | - | 23, 37 |
| $f_{lday}$ | Fractional day length | - | 3 |
| $f_{pe,red}$ | Fractional storage reduction due to permafrost | - | 30, 31 |
| $f_{pe}$ | Permafrost fraction | - | 30 |
| $f_{so,bs,low}$ | $S_{so}$ fractional lower limit for $E_{bs}$ | - | 22 |
| $f_{so,crit}$ | $S_{so}$ fraction for unconstraint transpiration | - | 21 |
| $f_{sw}$ | Surface water fraction | flake | 23 |
| $f_{veg}$ | Vegetation fraction | fveg | 7, 12, 13 |
| $f_{wetland}$ | Wetland fraction | - | 23, 37 |
| $f_{wet}$ | Wet surface fraction for skin and canopy | - | 8, 9 |
| $frc_{liquid}$ | Maximum liquid snow water fraction | - | 4 |
| $F_{snlq}$ | Refreezing snow melt | - | 4, 5, 6 |
| $\text{LAG}_{gw}$ | Shallow groundwater retention time | - | 25, 36 |
| $\text{LAG}_{rf}$ | River flow retention time | - | 27, 35, 38, 41 |
| $\text{LAG}_{sw}$ | Surface water retention time | - | 25, 35, 38, 41 |
| LAI | Leaf area index | - | 11 |
| $P$ | Total precipitation | precip | 2 |
| $PET$ | Potential evapotranspiration | potevap | 9, 21, 22, 23 |
| $P_{ra}$ | Rainfall | rainf | 7, 8, 9, 10, 23 |
| $P_{sn}$ | Snowfall | snowf | 1, 2, 3 |
| $Q_{rf,0}$ | River inflow from upstream cells | rivdis | 28 |

| Symbol | Description | Output Name | Equation |
|---|---|---|---|
| $Q_{rf,n}$ | River storage outflow from cascade n | - | 26, 27, 29 |
| $Q_{rf}$ | River discharge | dis | 28, 29 |
| $R_{can}$ | Runoff from canopy reservoir | - | 10, 12 |
| $R_{gr,high}$ | Highflow drainage component | - | 15, 16 |
| $R_{gr,low}$ | Lowflow drainage component | - | 14, 16 |
| $R_{gr,max}$ | Maximum drainage | - | 15 |
| $R_{gr,min}$ | Minimum drainage | - | 14, 15 |
| $R_{gr}$ | Subsurface drainage / groundwater recharge | qsb | 13, 16, 24, 25 |
| $R_{gw}$ | Grundwater runoff | qg | 24, 29 |
| $R_{skin}$ | Runoff from skin reservoir | - | 10, 12 |
| $R_{sn}$ | Snowmelt | smelt | 1, 3, 5, 7, 8, 9, 10, 23 |
| $R_{srf}$ | Surface runoff | qs | 13, 18, 23, 29 |
| $R_{sw}$ | Overland flow into river | qsl | 23, 25 |
| $R_{tr}$ | Throughfall onto ground | throu | 12, 13, 18, 20 |
| $\sigma_0$ | Minimum orographical sub-grid standard deviation | - | 32 |
| $\sigma_h$ | Orographical sub-grid standard deviation | - | 32 |
| $\sigma_{max}$ | Maximum orographical sub-grid standard deviation | - | 32 |
| $S_{can,max}$ | Maximum actual canopy storage | - | 8, 10, 11 |
| $S_{can}$ | Canopy reservoir | canopystor | 7 |
| $S_{gw}$ | Groundwater storage | groundwstor | 24, 25 |
| $S_{pe,max}$ | Maximum soil moisture in permafrost regions | - | 30 |
| $S_{rf}$ | Riverflow storage | riverstor | 26, 27 |
| $S_{skin,max}$ | Maximum actual skin storage | - | 8, 10, 11 |
| $S_{skin}$ | Skin reservoir | skinstor | 7 |
| $S_{snlq}$ | Snow liquid water content | wliq | 1, 4, 6 |
| $S_{sn}$ | Snow water equivalent | swe | 1, 3, 4 |
| $S_{so,max}$ | Maximum root zone soil moisture capacity | - | 14, 15, 16, 17, 18, 21, 22, 30, 31 |
| $S_{so,sg,max}$ | Maximum sub-grid soil moisture capacity | - | 17, 18, 19, 20, 31 |
| $S_{so,sg,min}$ | Minimum sub-grid soil moisture capacity | - | 17, 18, 19, 20, 31 |
| $S_{so,sg}$ | Sub-grid soil moisture capacity | - | 17, 18, 19, 20 |
| $S_{so,wilt}$ | Root zone moisture at the wilting point | - | 21, 31 |
| $S_{so}$ | Root zone soil moisture | rootmoist | 13, 14, 15, 16, 17, 18, 21, 22 |
| $S_{sw}$ | Surface water storage | lakestor | 23, 25 |
| $T_{melt}$ | Snow melt temperature threshold | - | 3 |

725

| Symbol | Description | Output Name | Equation |
| --- | --- | --- | --- |
| $T_{sn,min}$ | Minimum temperature for rainfall | - | 2 |
| $T_{sn_max}$ | Maximum temperature for snowfall | - | 2 |
| $T_{srf}$ | Surface temperature | tsurf | 1, 2, 3, 16, 18 |
| $v$ | Slope dependend water flow velocity | - | 34 |
| $v_{gw}$ | Water flow velocity for groundwater | - | 36 |
| $v_{rf}$ | Water flow velocity in rivers | - | 35, 38, 39 |
| $v_{sw}$ | Water flow velocity for surface water | - | 35, 38, 39 |

**Table 2.** Land surface data variables and their use (directly or as input for derived variables) in the default setup of HydroPy. A symbol is provided for all variables that are used in equations in the hydrological process (Sect. 2.2) or input data description (Sect. 3).

| Variable | Long Name | Symbol | Source | Utilization |
|---|---|---|---|---|
| lsm | Land sea mask | - | Hagemann (2002) | defines land cells |
| area | Grid cell surface area | $A_{gc}$ | Computed | unit conversions |
| glacier | Glacier fraction | - | Hagemann (2002) | excluded from simulation |
| perm | Permafrost fraction | $f_{pe}$ | GEWEX ISLSCP Project (2007) | Soil scheme |
| flake | Lake fraction | $f_{lake}$ | Lehner and Döll (2004) | Lake scheme, River routing |
| fwetl | Wetland fraction | $f_{wetland}$ | Lehner and Döll (2004) | River routing |
| fveg | Vegetated fraction | $f_{veg}$ | Hagemann (2002) | Canopy scheme, Soil scheme |
| lai | Leaf area index | LAI | Hagemann (2002) | Canopy scheme |
| srftopo | Mean surface orography | $h$ | Amante and Eakins (2009) | River routing |
| topo_std | STD of surface orography | $\sigma_h$ | Amante and Eakins (2009) | Soil scheme, Groundwater scheme |
| slope_avg | Slope | $s$ | Amante and Eakins (2009) | Lake scheme |
| wcap | Maximum soil water capacity | $S_{so,max}$ | Hagemann and Stacke (2015) | Soil scheme |
| wava | Plant available water | $S_{so,wilt}$ | Hagemann and Stacke (2015) | Soil scheme |
| wmin | Minimum sub-grid soil moisture capacity | $S_{so,sg,min}$ | Hagemann and Gates (2003) | Soil scheme |
| wmax | Maximum sub-grid soil moisture capacity | $S_{so,sg,max}$ | Hagemann and Gates (2003) | Soil scheme |
| beta | Distribution of sub-grid soil moisture capacities | $b_{sg}$ | Hagemann and Gates (2003) | Soil scheme |
| rout_lat | River flow target indices for latitudes | - | Hagemann and Dümenil (1997) | River routing |
| rout_lon | River flow target indices for longitudes | - | Hagemann and Dümenil (1997) | River routing |

**Table 3.** Reference values for lateral water flow computation obtained from sensitivity simulations described in Hagemann and Dümenil (1997).

| | $\mathrm{LAG}_{ref}$ | $n_{ref}$ | $v_{ref}$ | $\Delta l$ |
|---|---|---|---|---|
| Surface water | 50.5566 | 1.11070 | 1.0885 | 171000 |
| River | 0.41120 | 5.47872 | 1.0039 | 228000 |
| Groundwater | 300 | 1 | - | 50000 |