# Peer review of "HydroPy (v1.0): A new global hydrology model written in Python"

_Geoscientific Model Development, 2021_

## Author Response (AR1)

**Responses to reviewer comments**

We thank the reviewers for the critical and constructive discussions on our manuscript. Here, we will repeat the reviewer's comments (italic font) and response directly below (standard font).

**RC1**

*The model description paper "HydroPy (v1.0): A new global hydrology model written in Python" describes a reimplementation of the global hydrological model MPI-HM.*

*Overall, I certainly support the publication of this study; however, many questions remain unanswered, requiring a major revision of the current manuscript.*

*First of all, I think it is tremendous that the authors approached the challenge of rewriting a scientific software! A lot of software used in the community is stuck in the last century. Such an endeavour, reimplementing a research software, requires work that generally does not result in any measurable new science but still is extremely important and currently undervalued. While the authors made some reasonable choices in technology (choice of language and libraries), they completely ignore the work that has been done on other modelling projects. Furthermore, they claim benefits of the reimplementation that are not supported by any evaluation.*

We thank you very much for the appreciation of our work. We fully agree to the challenges you mentioned that were posed by this project. As the critical comments are repeated in the detailed comments section, we will discuss them there.

*Detailed Comments:*

*The title struck me as very odd. There are other global models as well that are written in Python, for example, PCR-GLOBWB; which also has been described in a model description paper in this very journal and has not been cited! In general, much relevant literature on global hydrological models is ignored. Indeed, this is not supposed to be a review paper but only citing your own modelling papers is not something that acknowledges the work of others and advances science. This also relates to the evaluation, which also did not include any comparison whatsoever to other models. At least the differences should be discussed if a full out comparison is not possible or due to calibration limited.*

With the choice of title we did not intent to imply that our model would be the only one written in Python, but rather wanted to provide context for our choice of the model name. For this reason, we made no dedicated effort to find and review other GHMs that use Python. Nonetheless, we will add a short paragraph on the general state of model development to the documentation and think about options to rephrase our title.

We like to emphasize, that we did not limit ourselves to citing only our own modeling papers. In most of the cases, we provided references to studies, which resulted from international inter-comparison projects (WATCH and ISIMIP) where a large number of GHMs, including the MPI-HM, took part. The cited papers also provide plenty of information about the scientific performance of the MPI-HM compared to other GHMs. Considering that HydroPy is not a new model in terms of its hydrological equations but a remake of the MPI-HM, we do not plan to repeat these comparisons with other models

at this stage. Instead, we demonstrated HydroPy's similarity with MPI-HM, which implicitly shows that the evaluations done for MPI-HM are still valid for HydroPy. In addition, we evaluated HydroPy against observational data, because this was not yet done for MPI-HM in previous publications. Of course, we will modify the text to make readers aware of the available model evaluation studies on MPI-HM. However, we feel that exercises like standardized model inter-comparisons for complex models like GHMs are a scientific project by themselves and beyond the scope of a model description paper.

*As already mentioned, I think a significant contribution is the reimplementation of a model. Sadly the authors only included minimal information and no discussion on the involved challenges etc. This would probably merit its own perspective piece in GMD, but still, it would benefit the community greatly if you would provide more insights on that process. Also, you should consider citing "Muller et al. Going open-source with a model dinosaur and establishing model evaluation standards, EGU 2018"*

Thanks for proposing a section on the challenges encountered during this project. This is a very good idea and we will add a paragraph on this to our manuscript. In summary, we find that such efforts are difficult as usually no (third party) funding is available for modernizing a scientific software. This leads to a lack of resources, especially of personnel with the required skill set and dedicated working time, and limits the model development perspective to the end of a scientist's contract. Furthermore, there is a difference between a 'good' model from the perspectives of scientific researchers and software engineers, as there has to be a balance between how easy model routines can be understood and modified versus the pure technical model efficiency in terms of computational resources and scalability. And finally, as the reviewer already mentioned, the direct scientific gain is rather low although publication of such effort in journals like GMD finally allow the scientists to earn some merits with technical work.

*It is great that the authors published their code as OpenSource, but I don't quite understand why the authors choose to upload the code to Zenodo but not to a platform like github or bitbucket. Or is it available there as well? Then please add a link the code availability section. It would benefit the community greatly if the code and its further development process are more accessible. This is, of course, nothing that should influence a decision on if this manuscript should be published but still something worth noting.*

No, we published HydroPy only on Zenodo. The reason is directly related to our answer to remark 2. Zenodo can be used as a repository for source code (and data) and even allows to add new versions whenever we are able to work on model improvements. This suits our main interest, which is to support transparency in science by making our source code public. In contrast, platforms like github are more tailored towards interactive model/software development together with the respective community. Of course, we are aware that such collaboration can improve software significantly, as very skilled people might join the effort. However, our limited resources do not allow for any reliable model support beyond providing a manual with setup information. Furthermore, the direction of model development for HydroPy is determined by the projects we can get funding for and not by the needs and wishes of the community. While this situation is not satisfying for us either, we feel that publication on Zenodo enables us to make most of our limited resources.

*In the abstract, the authors claim that "the new model requires much less effort in maintenance and due to its flexible infrastructure, new processes can be easily implemented". I do not see any evidence for these claims. While the code is thoroughly documented, files with over a thousand code lines and 3-4 classes do not reflect a carefully designed software architecture with extensibility and maintainability in mind. The paper is not a computer science manuscript, and thus I have to credit that they provided a rather clean implementation. Yet, the authors need to either compute metrics that support their claims or discuss their software architecture in more detail explaining how it supports the integration of new processes. Did you use particular software patterns to ensure that? Or do you just hope to achieve that because you used a more modern language? I do not want to discredit the tremendous work that probably went into the implementation, but such claims need to be supported; otherwise, it is not much of a scientific publication. Furthermore, I urge the authors to take a look at the review guidelines of JOSS to improve their code further. Currently, you don't have any automated tests that would allow a 3rd party, and more importantly, you, to check if your software is working correctly. Again this is not a criterion that GMD is added to its guidelines and will not prevent me from supporting a publication.*

You are right: these claims are not supported in the manuscript. Actually, our statement concerning the infrastructure and model design were not meant in an objective way, but rather subjectively in comparison to the model structure and setup of the MPI-HM model. We fully acknowledge that there are probably concepts around to improve the structure of the code and its maintainability. However (see response to remark 2), we do not have a software engineer in our team and, thus, are limited to the scientist's point of view. Nonetheless, we agree that some information should be added to the manuscript. We are not aware of metrics to objectively evaluate the quality of our model design, but we will add a paragraph on our reasoning behind the code structure and our intended workflow to add new processes.

*Please add the central variables to figure 1. It would help significantly understand how the implemented processes work together and keep an overview of all the variables.*

We deliberately omitted this information from the figure as we wanted to present a general overview over the hydrological processes realized in the model. Adding specific variables to the figure requires much more (and therefore smaller) text and interaction descriptions. However, we will try to add as much of these information as possible without making the figure too confusing.

*Please add a table of all variables, a short explanation and in which equation they are used and if they are available as output.*

We will add such a table.

*You mention the land surface property data in the data availability section but not the output data used in the evaluations section of the model. Please make this available or state why it is not possible. See also a recent paper in GMD as a possible example: https://gmd.copernicus.org/articles/14/1037/2021/*

At the point of submission, we were not convinced that the raw model output is of major interest to the community and did not want to potentially waste storage on any public repository. Of course, we are happy to follow your request and we will add a link to our manuscript.

*Line 71: How is f_snlq,max determined?*

The value of 6% for maximum liquid snow water content is commonly used e.g. in Wigmosta et al 1994. We will add this information to the manuscript.

*Line 95: Please discuss the implications of not considering groundwater recharge (defuse and focused)*

This might be a misunderstanding, thanks for making us aware of this. In our simple model structure, groundwater recharge R_gr is set equal to drainage and subsurface runoff and is used in the balance equations of soil moisture (Eq. 13) and shallow groundwater (Eq. 24). We will add groundwater recharge to the list of synonyms for drainage.

Line 139: Unclear, please elaborate

In MPI-HM the vertical land surface water balance module and the routing module were strongly separated, each featuring storages which were restricted to be used within the respective module. The sole exception was the surface water storage resulting from ponding water on the land surface, and, hence, used to to represent small creeks, lakes and wetlands. This storage was part of the vertical land surface water balance but could also interact with the lateral river routing scheme. In HydroPy, we simplified the representation of lakes and wetlands and utilize storages that are already part of the routing scheme.

*Line 175 ff: Why not as inline c code and compile with cython? That would make it more accessible. In general, could you provide some performance metrics? Is the new implementation much faster/slower?*

Prior to using fortran, we experimented with a cython implementation. However, the increase in performance was not as high as we hoped for and, thus, we switched to a fortran version. We tested this with simulations done on an office laptop (Intel Core i5 9400H with 8GB Ram). Without writing any output, simulations at 0.5 deg on a global domain for 1 model month took:

| pure python | 22.3 s (+- 160ms) |
| python + cython routing | 18.9s (+- 111ms) |
| python + fortran routing | 13.6s (+- 38ms) |

Please note, this numbers are from an older model version and probably not up to date anymore. We will provide performance numbers for the pure python and fortran routing version in the revised manuscript.

*The whole comparison to MPI-HM is solely focused on the NSE. This does not prove that you are getting identical/similar results for the right reasons. Again automated tests would be a great addition. Furthermore, I greatly recommend a full sensitivity analysis using, for example, Morris.*

This is done, because river discharge is essentially the target variable of our GHM and it depends on all prior computations done in the model. Thus, focusing on river discharge to demonstrate the similarity

in model output between HydroPy and MPI-HM appears as the most logical choice to us. Of course, you are right that similar results could theoretically be achieved for the wrong reasons. However, considering that both models are based on the same physical hydrological equations, we think the likelihood for such a 'false positive' would be rather low. Additionally, while the (N)NSE admittedly plays a large role in our comparison of river discharge, it is by no means the sole focus. For river discharge we present climatological river discharge curves for catchments of interest to explain differences between both model setups (Fig. 11 in the manuscript) and before that we compare and discuss average water fluxes and storages which exist in both models (Fig. 9 in the manuscript).

Concerning automatic testing, we will keep this recommendation in mind. But as we don't have any software engineer available for such task, we expect the development of meaningful tests to take a certain amount of thinking and revision and won't start on this in the framework of the current study.

Similarly, we agree that sensitivity analyses like the Morris One-At-a-Time or Latin Hypercube are very useful tools to identify the most important parameters in a model and prioritize them concerning tuning, optimization or simplification. However, in this study we just want to do the first step, namely the re-implementation of the MPI-HM in a modern form as HydroPy. While improving on the original formulations is certainly on our agenda, it has to wait until our limited resources allow for this.

*You should discuss the performance of your model with respect to other available models.*

We are not sure whether you refer to the scientific performance or the technical one. Concerning scientific performance, we already commented on this in our reply to remark (1). Of course, we will include our reasoning about the implicit comparison due to the similarity to MPI-HM at the end of Sect. 4.4. Concerning the technical performance, we can provide numbers about model runtime, but we don't see much value in comparing those to other models. GHMs vary strongly in complexity and the number of implemented processes. Whether a model has a higher (computational) efficiency compared to another would be only relevant if both are equal in their outcome as well as scope of applications. Thus, we cannot include such a comparison in our manuscript and furthermore argue that such a project would be a major study on its own and beyond the scope of a model documentation paper.

**RC2**

*In this manuscript, the authors provide a detailed description and validation of the global hydrological model HydroPy. The model is based on an existing model (MPI-HM) but has been completely rewritten in Python and made publicly available (along with the appropriate input data) under a GNU GPL license. The paper is well written and clearly explains all relevant processes in the model. Strengths and shortcomings are discussed, and potential targets for future model improvement are identified.*

Thank you, we very much appreciate this positive evaluation of our work!

*Lines 111-113: "It assumes … storage overflow". These two sentences appear contradictory. Also, I have difficulties to relate them to the following technical description. Why is $S_{SO,sg}$ scaled because $S_{SO,max}$ is assumed to vary? This section would benefit from a more extensive description of the infiltration/runoff scheme and the concept behind it. In the end, this is one of the main components of the model.*

We will add some more information to this paragraph to make it less confusing. The new version states:

It assumes that $S_{so,max}$ is not homogeneously distributed within a grid cell, but varies on subgrid scale. Thus, parts of the grid cell where the local storage capacity is low, can already generate surface runoff even though the cell average soil moisture state is still below its average maximum moisture holding capacity. Therefore, a fraction of $R_{tr}$ is converted into $R_{srf}$ as soon as the minimum soil moisture content is exceeded. This is realized by mapping $S_{so}$ onto the sub-grid soil moisture capacity distribution parameters denoted by the index $_{sg}$.

*Lines 182-186: It is very prudent to include such a feature. However, I would consider letting the simulation fail or at least throw a warning if a balance error occurs. Not all potential users of the model may be aware that the balance needs regular checking.*

Indeed, we implemented the water balance check in such way that at the end of every simulation the global water balance components (both averaged over all grid cells and as global sum) are displayed and color coded in green and red to indicate a closed or violated water balance. Additionally, the data file including the water balance fields is only written in case the water balance is not closed and therefore it can be seen at once if any problems occurred and in which year this happened. We will extent the manuscript to include this information.

*Lines 206-207: "…wetlands were restricted to…". I am aware that GLWD identifies huge areas as wetlands in North America, which apparently doesn't correspond well to your model assumptions. But it feels a bit arbitrary to exclude the largest part of global wetlands just because you are unhappy with the results. What are the implications in other parts of the world? Since this is just a test setup, I see no urgent need to change this decision within the current study. But you should clearly flag it as a mismatch between the model and available data and come back to it in the discussion on wetlands in the section 4.4.*

That is a very valid remark and we agree that some more information on our reasoning should be provided. The basic idea is that (at this stage) we wanted HydroPy to reproduce MPI-HM simulations as best as possible to have a robust reference for all further development. At the same time, we found it important to update some of the rather old datasets used for the land surface characteristics. Our GLWD wetland class selection was not so much guided by optimizing the selection to get the best possible discharge, but rather to best resemble the wetland distribution in the Matthews and Fung dataset (used for MPI-HM) to allow for a fair comparison between both. We will modify the sentence to emphasize this reasoning:

The lake fractions are used without any modifications, but wetlands were restricted to the classes floodplains and peatlands for this study. Thus, we can best resemble the general wetland distribution used for the predecessor of HydroPy and facilitate a clean comparison between both (see Sect. 4.4).

*Lines 210-211: Note that such interpolation alters the effective monthly averages, with largest effects in months with minima and maxima. However, it is common practice and not easy to correct for.*

We fully agree with the reviewer. Still, we are confident that the error caused by this approach should not affect the simulations too much for the rather large time scales the model is intended for. Of course, any results for short term application would need to take this effect into account.

*Line 244: "…distribution parameter $b_{sg}$…". Can you give a value for this parameter?*

This parameter is a spatially distributed field provided in the land surface dataset, not a single value. For the majority of grid cells, the value is rather low (<5) but in more extreme (and only very few) cases is can reach values up to 100.

*Lines 249-281: Please assure consistent use of subscript x in text and equations.*

We are not sure whether we recognized the issue you are pointing at. Is it, because we are also using a subscript y in this paragraph? Here, it is necessary, because Eq 39 and 40 are 2-dimensional using the subscript x for flows in certain water body types (surface water and river) and y for the land cover types (lakes and wetlands) the water flows through. Anyway, we will check all subscripts for consistency and consider replacing generic subscripts wherever appropriate.

*Lines 286-287: "The simulation … until 2014". If I understand correctly, you are trying to bring your model to an equilibrium using conditions in a single year (1979). Wouldn't it be better to use a range of years or at least a climatology?*

Yes, our intention is to provide the model with a storage state where changes are dominated by climate input and not by any residual trends. There are several (small) advantages and disadvantages to the methods: using a climatology might reduce the day to day variability and thus limit the model's exposure to any daily extremes. Picking just one year (as we did) could lead to a bias in storage state in case the particular year happens to be an extreme year in a given region but allows for a very clean setup with a constant storage target, which the model approaches. Using a time series for spinup generates the exact opposite, e.g. extreme years and therefore an anomalous storage state are less likely. However, the target state is less well defined due to inter-annual variability. We decided to use the approach with the best defined target state because it allows for a more straight forward evaluation of the spinup behavior. Anyway, we claim that our model is not very sensitive to either method as long as the number of spin-up years and the general climate are similar. In order to test this hypothesis, we performed another spin-up simulation using a time series from 1930 until 1979 (instead of a single year) as well as another production simulation from 1979 until 2014 based on this spinup. We then compared our original simulation and the new simulation (Fig 1). This comparison demonstrates that indeed both methods cause differences in the production simulation. However, these differences are very low resulting in an RMSE which rarely exceeds 5 kg m-2 a-1. Furthermore, the long term trends found in both simulations are very similar with only a small numbers of grid cells that show different trends due to the different spinup methods. From this, we conclude that the choice of spinup method has only a minor effect on our results.

[Figure]

*Figure 1: Impact of different spinup methods on root zone soil moisture during the simulation period. The left panels show the spatial distribution of root zone moisture RMSE and differences in long term trends between simulations initialized with multi-year or single-year spinup. The right panel displays the relation between trends in the different simulations at grid cell scale. Note, that all panels displays root moisture and not TWS as the latter is affected by single cells of steadily increasing snow cover (see Sec 4.1 in the manuscript).*

Lines 306-320: I very much doubt that checking for the pixel with the largest absolute residual trend is sufficient proof that the model is in equilibrium. In many regions, the absolute storage trends are small because the involved water flows (precipitation, runoff, recharge, discharge) are small. Thus, storages (and fluxes) will take much longer to reach equilibrium in regions where initial storage trends are small (a fairly large part of the world, according to the map in Figure 3). This is of minor concern when considering globally aggregated fluxes, which are dominated by regions with large fluxes. But I suspect that many of the other results presented in the remainder of the paper are affected by storages not being in equilibrium at the beginning of the simulation phase. I strongly suggest using a different indicator for quantifying the proximity to equilibrium and adjusting the spin-up protocol accordingly.

We used this method exactly for the reason you mentioned: any residual trend is most likely only found in regions where the annual water turnover is much smaller than the storage capacity. However, we think such regions are very few and do not interfere with any of our analysis, either on global nor on catchment scale. To some part this is already demonstrated in Fig 1 (right panel) which shows the similarity of simulated long-term trends in spite of using different spin-up methods. Additionally, you can see in Fig. 2 the trends in total water storage for the last 10 years of spinup up simulation. Apart from the already discussed grid cells in glaciated regions, there are very few grid cells which show any residual trend (still being rather low with < 5 kg m-2 a-1), but these are located in rather wet regions like the Ganges Delta and the Pantanal wetlands.

[Figure]

*Figure 2: Spatial distribution of yearly trends during the last 10 cycles of the spinup simulation.*

Based on this analysis, we are confident that none of our results are significantly affected by spinup. We will extent Fig 3 of our manuscript to include this plot as well.

*Line 370: "…temporal correlation…". What time step is used here?*

The analysis time step is monthly (because we have only monthly data available for observations). We will add this information.

*Line 374: What do you mean by "percentile bias…"?*

The percentile bias is the ratio of the sum of differences between simulation and observation to the sum of observed values given as percentile. We will add the equation to the sections.

*Line 416: What do you mean by "mitigated flow curve"?*

We mean that the flow curve has a delayed and lower peak than its reference. As we say this in the next part of the sentence anyway, we will remove 'mitigated' from the sentence, to avoid any confusion.

*Lines 439-441: This would also solve the problem that led to the exclusion of wetlands, correct?*

Yes, we think this would very likely be the case.

**Overview on manuscript changes**

In response to the reviewer's comments we modified several parts of the manuscript:

- extended the introduction to provide additional information about the state of development for GHMs

- moved information about the model time loop to the beginning of the model description section

- added a section providing information on model design, structure, setup and technical performance

- revised model description section to clarify some statements (e.g. surface runoff, shallow groundwater, equation indices

- added equations for correlation and percentile bias computation

- added a paragraph on challenges and advice related to rewriting a complex modeling

- added data availability statement for our raw simulation data. The data review for upload to the CERA database is not yet finished but a DOI for public access is expected to be available prior to the final publication.

- Revised Fig. 1 to include the major variables (+ added a logo for HydroPy)

- Added two more Figures (2 and 3) to visualize the model structure and performance

- Modified Fig 5 (former Fig 3) to demonstrate that all states are in equilibrium at the start of the simulations

- Added a new table (Table 1) containing the list of variables used in the model description section, their meaning, output availability and numbers of equation where they are used

- Modified Table 2 (former Table 1) to include the symbol for variables used in equations throughout the manuscript

---

## Author Response (AR2)

**Responses to reviewer comments**

We thank the reviewers for spending the time for another review of our manuscript. Like last time, we will repeat the reviewer's comments (italic font) and response directly below (standard font).

**RC1**

*The authors have revised the manuscript according to the reviewers' comments. They have also updated and extended text and figures to further improve the manuscript. I think the manuscript has developed a lot and I find most of my comments well addressed.*

Thank you.

*However, I still don't agree with the way the authors evaluate whether water storages have reached equilibrium in their model. In dry regions, where inflows, outflows, and equilibrium storage are small, a small residual trend in storage change does not necessarily indicate that storages have reached equilibrium. An example for a more suitable metric would be the ratio of residual storage change to the sum of the storage change and the storage outflow (which should equal the inflow). This provides a direct measure for how much of the inflow is still used to fill up the storage and, by that, by how much outflow is still affected. I understand that for the current study, it is not very critical to have all water storages in equilibrium. But since the authors have dedicated a relatively large portion of the paper on this issue, it would seem important to address it appropriately. I would like to see the shortcomings of assessing equilibrium based on absolute residual storage changes at least briefly discussed. Perhaps the authors could complement that by an estimate of how many grid cells are in equilibrium using the metric above, for example.*

We assume that part of our disagreement on this point might be due to a different perception of the goal of this spin-up evaluation. Our reasoning is not to provide the model with a perfect initial state. Actually, this would not be the best initialization anyway, as many regions (e.g. desiccating lakes) are not in an equilibrium state in reality. Thus, the initial state we strive for is one that the model can use without:

a) experiencing any kind of initialization shock due to large changes in storages during the first steps of the simulation which might interfere with the results afterwards and

b) experiencing any residual trends which might then be wrongly mistaken for real signals.

For this reason, the state of very dry grid cells do not matter as much for our simulations (as the reviewer already acknowledged). Moreover, for the same reason, we indeed think that trends provide a good measure of the suitability of our initial state because their size and distribution tells us, whether any residual signals might be expected in our production simulation.

Nonetheless, we very much thank the reviewer for proposing this interesting alternative metric. Applying it for the last 10 years of our spin up simulation confirms our spin-up evaluation using trends. With the exception of the mentioned glacier cells where the spin-up fraction is still above 10%, there are less than 100 cells that show a residual storage change larger than 0.1% of the annual sum of outflow and storage change.

[Figure]

Furthermore, these cells are not located in dry areas, but rather along main river channels. Exactly the same signal pattern occurs in our trend analysis. However, they correspond to trends <= 0.1 kg m-2 a-1 and, thus, do not show up in Fig. 5 of the manuscript as we consider such trends to be negligible for our simulation and chose our color map accordingly.

Although we already mentioned in our manuscript that the spin-up target is the production of a suitable field for model initialization and not a real equilibrium state, we further modified the text and replaced all mentions of "equilibrium" with "stable state" to better reflect our intention with this analysis.

*One other point I didn't catch in my first review is the interpretation of model performance based on normalized Nash-Sutcliffe Efficiency (NNSE). The NNSE range of what is considered sufficient performance in Moriasi et al. (2007) refers to daily discharge time series. Applying these thresholds to NNSE calculated for monthly discharge climatologies is inappropriate and falsely implies a performance similar to calibrated watershed models. I think a clarification of the how NNSE values of monthly discharge climatologies are to be interpreted is needed here.*

After carefully re-checking the study of Moriasi et al (2007), we do have to disagree on this point. Table 4 (page 891 in Moriasi et al. 2007), which is the source of our values, is even explicitly named: "General performance ratings for recommended statistics for a monthly time step." On the same page, it is mentioned that "The model evaluation guidelines presented in the previous section apply to the typical case of continuous, long-term simulation for a monthly time step." For this reason, we don't think that we falsely imply a very good performance. Especially, as we don't claim it globally, but just for a minority of catchments. Please note, that while working on your remarks, we actually found a bug in our NSE calculation, however, the 20% best-performing catchments were hardly affected by it.

Anyway, as there is a general trend to prefer the Kling-Gupta efficiency (KGE) over NSE (e.g. Knoben et al, 2019), we meanwhile changed our analysis setup to use the KGE instead and adapted our analysis accordingly. Thus, we updated those parts of our analysis that were based on the NSE (Sec. 4.3 and Sec 4.4), with the corresponding figures 8, 12 and 13. We also slightly modified the selection of river basins to discuss the differences between HydroPy and MPI-HM. Contrary to the NSE, there are no specific categories defined for the KGE but generally positive values are considered to indicate model skill (Knoben et al, 2019). For this reason, we removed the sentence comparing HydroPy to calibrated

catchment models. Note, that the switch from NSE to KGE does not change any of the conclusions of this study.

---

## Author Response (AR3)

Addon to the Author's certification about model content

The submitted files are identical to the version accepted by the topical editor except:
- I removed one reference (Line 501: Kumar et al., 2021) from a list of examples, as it's current status is unclear (might not be published) and being just one of several examples, it is not required
- I updated the reference lists concerning accepted papers and journal abbreviations